# CONFORMAL TRAINING WITH REDUCED VARIANCE

## ABSTRACT

Conformal prediction (CP) is a distribution-free framework for achieving probabilistic guarantees on black-box models. CP is generally applied to a model post-training. Recent research efforts, on the other hand, have focused on optimizing CP efficiency *during training*. We formalize this concept as the problem of *conformal risk minimization* (CRM). In this direction, conformal training (`ConfTr`) by Stutz et al. (2022) is a technique that seeks to minimize the expected prediction set size of a model by simulating CP in-between training updates. Despite its potential, we identify a strong source of sample inefficiency in `ConfTr` that leads to overly noisy estimated gradients, introducing training instability and limiting practical use. To address this challenge, we propose *variance-reduced conformal training* (`VR-ConfTr`), a CRM method that incorporates a novel variance reduction technique in the gradient estimation of the `ConfTr` objective function. Through extensive experiments on various benchmark datasets, we demonstrate that `VR-ConfTr` consistently achieves faster convergence and smaller prediction sets compared to baselines.

## 1 INTRODUCTION

Consider a classification task with input (features) $X \in \mathcal{X}$ and corresponding label $Y \in \mathcal{Y} = \{1, \ldots, K\}$. In supervised learning, we are interested in approximating the posterior probability $\pi(y|x) = \mathbb{P}\left(Y = y \mid X = x\right)$ by tuning some $\theta$-parameterized family of predictors $\pi_\theta(y|x)$ - for example, neural network models with a softmax activation at the output layer. Typically, the final label prediction would be $\delta_\theta(x) = \arg\max_{y \in \mathcal{Y}} \pi_\theta(y|x)$, and a common metric for performance is the *accuracy*, which measures the proportion of testing samples whose predicted label matches the true label. While the accuracy is a key performance metric, in safety-critical applications with a downstream decision maker it is crucial not only to predict accurately but also to quantify the uncertainty associated with a prediction.

Conformal prediction (CP) is a distribution-free, principled framework that is used to provide formal probabilistic guarantees for black-box models (Vovk et al. (2005); Shafer & Vovk (2008); Angelopoulos et al. (2023)), with exemplar applications in computer vision (Angelopoulos et al. (2020)), large language models (Mohri & Hashimoto (2024),Kumar et al. (2023)) and path planning (Lindemann et al. (2023)). Given a model $\pi_\theta(y|x)$, CP enables the construction of set predictors $C_\theta : \mathcal{X} \to 2^{\mathcal{Y}}$ (where $2^{\mathcal{Y}}$ is the power set of $\mathcal{Y}$) such that the true label is contained in the set of predicted labels with high probability. This can be formalized via the notion of *marginal coverage*.

**Definition 1.1** (Marginal coverage). We say that a set predictor $C_\theta : \mathcal{X} \to 2^{\mathcal{Y}}$ satisfies *marginal coverage* with miscoverage rate $\alpha \in (0, 1)$ if $\mathbb{P}\left(Y \in C_\theta(X)\right) \geq 1 - \alpha$.

Marginal coverage can be readily obtained in CP via a process called *calibration*, which only requires access to a so-called *calibration set* of data that is statistically exchangeable with the test data. However, one of the main challenges in CP is the *efficiency* of the prediction sets - namely the size of the sets $C_\theta(x)$ - often referred to as *length efficiency* (Fontana et al., 2023). For instance, while it is possible to trivially achieve the desired coverage by including the entire label space in $C_\theta(x)$, such an approach results in non-informative and excessively large prediction sets. An efficient $C_\theta(x)$ is as small as possible while still maintaining the coverage guarantee.

Various existing approaches, including the works by Romano et al. (2020); Yang & Kuchibhotla (2024); Bai et al. (2022), address the efficiency challenge by refining the conformal prediction procedure applied post-training to a black-box model. These methods, though effective, are constrained

by the performance of the pre-trained model $\pi_\theta(y|x)$ on which they are applied. On the other hand, recent research efforts (Dheur & Taieb, 2024; Cherian et al., 2024; Einbinder et al., 2022; Stutz et al., 2022; Bellotti, 2021) have focused on integrating CP directly into the training process. This provides a way to improve the CP efficiency also in the model optimization phase - when learning the parameter of the models - enabling a higher degree of control over the probabilistic guarantees efficiency. In this work, we formulate this approach as *conformal risk minimization* (CRM) and we focus on CRM for length efficiency optimization. We consider a setting similar to Stutz et al. (2022), who proposed conformal training (`ConfTr`), an algorithm achieving promising performance in improving the length-efficiency of the prediction sets constructed post-training.

Despite encouraging preliminary results, `ConfTr` introduces significant optimization challenges, particularly due to the use of differentiable approximations of CP sets. Indeed, `ConfTr` requires differentiating a loss function obtained simulating CP during training. This, in turn, requires accurately estimating the population quantile of the conformity scores and its gradient, which can be difficult with the limited data available in each mini-batch. Hence, training can exhibit large fluctuations in the loss and slow convergence, thereby reducing the practical applicability of the method.

In this work, we address these challenges by introducing *variance-reduced conformal training* (`VR-ConfTr`), a novel CRM algorithm leveraging a variance reduction technique for the estimation of quantiles' gradients. Relative to `confTr`, our proposed `VR-ConfTr` solution significantly stabilizes training - leading to faster convergence, and consistently enhances the length efficiency of post-training conformal prediction sets when compared against baselines.

## 1.1 CONTRIBUTIONS

Our contributions can be summarized as follows:

**Conformal risk minimization.** We formulate *conformal risk minimization* (CRM) as a framework for training a parameterized predictor that learns according to CP efficiency metrics.

**A "plug-in" algorithm.** Focusing on CRM for length efficiency optimization, we provide a novel analysis for the variance of the `ConfTr` (Stutz et al., 2022) method, which shows the need for improved estimators of the quantile gradients. Then, we introduce the pipeline of variance-reduced conformal training (`VR-ConfTr`), our proposed algorithm to overcome this challenge, which leverages a "plug-in" step to incorporate improved estimates of quantiles' gradients in the training.

**Novel variance reduction technique.** Building on a fundamental result, which characterizes the gradient of the population quantile as a conditional expectation, we propose a novel estimator for quantile gradients whose variance is provably reduced with the training batch size. This novel estimator can be seamlessly integrated into `VR-ConfTr`. We analyze the bias-variance trade-off of this novel estimator and establish its precise relationship with the conformity measures associated to a predictor $\pi_\theta(y|x)$.

**Empirical validations.** We extensively analyze our method on various benchmark and real-world datasets, including MNIST, FMNIST, KMNIST and OrganAMNIST. Our results demonstrate that `VR-ConfTr` consistently and significantly improves the efficiency and stability of conformal training for length efficiency optimization.

**Broad applicability.** Our approach and novel variance reduction technique can be integrated into any CRM method that requires quantile gradient estimation, extending its utility to a large class of conformal prediction frameworks and learning models.

## 1.2 RELATED WORK

A large body of research has focused on optimizing length-efficiency in CP. We now review some recent literature in this area. We first (i) review approaches that apply CP post-training to black-box models, and then (ii) review the recent efforts in coupling CP and model training, in what we call *conformal risk minimization* (CRM) approaches. For (i), recent algorithmic developments address improving length efficiency through better **conformity score** design Romano et al. (2020); Yang & Kuchibhotla (2024); Amoukou & Brunel (2023); Deutschmann et al. (2024); Luo & Zhou (2024). From another perspective Kiyani et al. (2024); Bai et al. (2022); Yang & Kuchibhotla (2021); Colombo & Vovk (2020) focus on designing better **calibration procedures**. Particularly, Kiyani

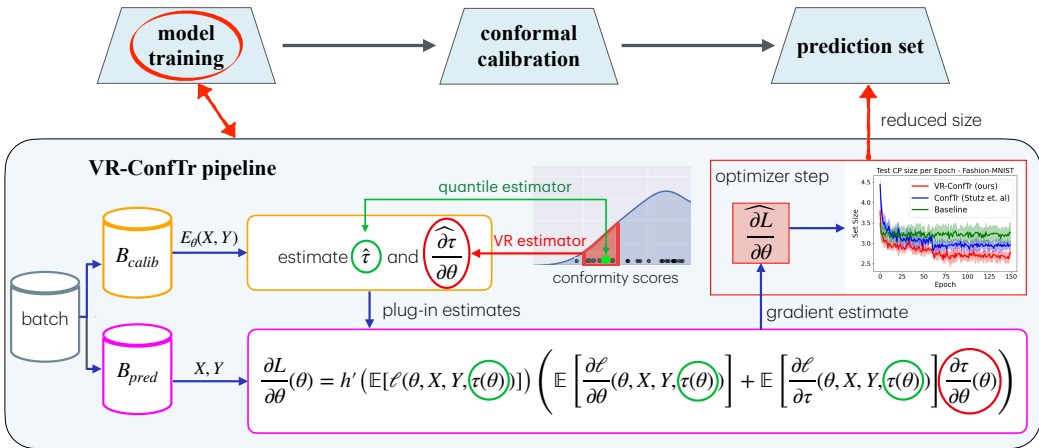

Figure 1: In this figure, we illustrate the `VR-ConfTr` pipeline and position it with respect to a typical CP procedure.

et al. (2024) propose an optimization technique for the calibration threshold $\tau$. On the other hand, Bai et al. (2022); Yang & Kuchibhotla (2021) formulate CP as a constrained optimization problem, minimizing the average prediction interval width with valid empirical coverage. These efforts do not fall under the CRM framework because they focus on learning low-dimensional hyper-parameters for pre-trained models as opposed to fully guiding the training of the parameters $\theta$ of a model $\pi_\theta(y|x)$.

**Conformal risk minimization.** There is a growing body of work (Einbinder et al., 2022; Cherian et al., 2024; Stutz et al., 2022; Bellotti, 2021; Yan et al., 2024) integrating ideas from conformal prediction in order to directly train a model for improved uncertainty quantification. Cherian et al. (2024) train a score function, rather than a point predictor, subject to conditional coverage constraints (Gibbs & Candes (2021)). Einbinder et al. (2022) utilize conformal prediction insights in order to mitigate overconfidence in multi-class classifiers by minimizing a carefully designed loss function. Stutz et al. (2022) proposed conformal training (`ConfTr`), in which length efficiency optimization is tackled by defining a loss function obtained by simulating conformal prediction during training. We will extensively describe this approach in the next section. Yan et al. (2024) uses a similar training pipeline to Stutz et al. (2022) in order to minimize the inefficiency of their proposed conformal predictor. The early work by Bellotti (2021) considered an approach analogous to `ConfTr` in that the authors simulate conformal prediction during training. However, the algorithm provided by Bellotti (2021) treats the quantile-threshold as fixed and not as a function of the model parameters, and it has been extensively shown by Stutz et al. (2022) that this approach provides inferior performance with respect to `ConfTr`.

## 2 PROBLEM FORMULATION

Let us consider a parameterized model of logits $f_\theta : \mathcal{X} \to \mathbb{R}^K$ and let $\pi_\theta(x) = \text{softmax}(f_\theta(x))$ denote the corresponding predicted probabilities. The objective of a conformal prediction algorithm is to construct a set predictor $C_\theta : \mathcal{X} \to 2^{\mathcal{Y}}$ starting from the model $f_\theta$ in such a way that $C_\theta$ achieves marginal coverage. One common way to achieve this is via a *thresholding* (THR) set predictor (Vovk et al. (2005)), $C_\theta(x; \tau) = \{y \in \mathcal{Y} : E_\theta(x, y) \geq \tau\}$ for some well chosen threshold $\tau$ and *conformity score* $E_\theta(x, y)$, which can be any heuristic notion of uncertainty regarding label $y$ upon input $x$ for the predictor $f_\theta(\cdot)$. Some choices for the conformity score include *(i)* the predicted probabilities $E_\theta(x, y) = \pi_\theta(y|x) = [\pi_\theta(x)]_y$, *(ii)* the logits $E_\theta(x, y) = [f_\theta(x)]_y$, and *(iii)* the predicted log-probabilities $E_\theta(x, y) = \log \pi_\theta(y|x)$. Let us assume that $X$ is an absolutely continuous random vector. If we knew the marginal distribution for $(X, Y)$, then marginal coverage could be readily achieved by setting $\tau = \tau(\theta) = Q_\alpha(E_\theta(X, Y))$ where $Q_\alpha$ denotes the population quantile of some scalar random variable. Indeed,

$$\mathbb{P}\left(Y \in C_\theta(X; \tau)\right) = \mathbb{P}\left(E_\theta(X, Y) \geq \tau\right) \geq 1 - \alpha \tag{1}$$

directly from $\tau = Q_\alpha(E_\theta(X, Y))$. In practice, however, the marginal distribution of $(X, Y)$ is not known. Instead, we can estimate $\tau(\theta) = Q_\alpha(E_\theta(X, Y))$ from data that, as long as it satisfies the principle of *exchangeability*, can be used to ascertain marginal coverage of $C_\theta(x; \tau)$.

## 2.1 CONFORMAL RISK MINIMIZATION

As we outlined in the introduction, recent research efforts have attempted to combine training and conformal prediction (CP) into one, as opposed to using CP only as a post-training method. Here, we formalize this by borrowing terminology from statistical supervised learning and by introducing the problem of *conformal risk minimization* (CRM). CRM can be understood as a framework for training a parameterized predictor that learns according to some CP efficiency metric, and can be formulated as follows:

$$\min_{\theta \in \Theta} \{L(\theta) := \mathbb{E}\left[\ell(C_\theta(X), Y)\right]\} \tag{CRM}$$

for some *conformal loss* $\ell$, where $C_\theta(x)$ is a *conformalized* predictor. This problem is closely related to the *conformal risk control* explored byAngelopoulos et al. (2022).

## 2.2 CONFTR (STUTZ ET AL., 2022)

Stutz et al. (2022) introduced conformal training (`ConfTr`), which we can categorize as a CRM approach for length efficiency optimization. In particular, `ConfTr` focuses on reducing *inefficiency* of calibrated classifiers, quantified by the *target size* of predicted sets. This can be understood as the problem in (CRM) with $\ell(C, y) = \max(0, |C| - \kappa)$ for some *target size* $\kappa$ (intended to discourage no predictions at all). In this regard, it is worth noting that the earlier work of Sadinle et al. (2019) was the first to study the closely related problem of *least ambiguous* set-valued classifiers, which corresponds to $l(C, y) = |C|$. An important aspect of the work of Stutz et al. (2022) is that the authors relaxed the CRM problem with target size conformal loss $\ell(C, y) = \max(0, |C| - \kappa)$ into a smooth one in $\theta$, in order to allow gradient-based optimization to be employed. In particular, the authors relax the prediction set $C_\theta(x; \tau)$ into the *smooth* prediction "set" (vector) $\mathbf{C}_\theta(x; \tau) \in [0, 1]^K$ with relaxed binary indicator variables, given by

$$[\mathbf{C}_\theta(x; \tau)]_y = \sigma\left(\frac{E_\theta(x, y) - \tau}{T}\right) \tag{2}$$

for $y \in \mathcal{Y}$, where $\sigma(\cdot)$ denotes the sigmoid function and $T > 0$ a "temperature" hyper-parameter such that $[\mathbf{C}_\theta(x; \tau)]_y \to 1_{E_\theta(x,y) \geq \tau}$ as $T \to 0$, with $1_A$ the indicator function for condition $A$. Further, the prediction set size $|C_\theta(x; \tau)|$ is relaxed into the smooth prediction set size $\sum_{y \in \mathcal{Y}}[\mathbf{C}_\theta(x; \tau)]_y = \mathbf{1}_K^\mathsf{T}\mathbf{C}_\theta(x; \tau)$. With this, the problem solved by Stutz et al. (2022) can be written as

$$\min_{\theta \in \Theta}\{L(\theta) = \log \mathbb{E}\left[\Omega(\mathbf{C}_\theta(X; \tau(\theta)))\right]\} \tag{3}$$

with $\Omega(\mathbf{C}) = \max(0, \mathbf{1}_K^\mathsf{T}\mathbf{C} - \kappa)$. Additionally, the authors explored other terms, such as a configurable class-conditional "coverage loss"

$$\mathcal{L}(\mathbf{C}, y) = \sum_{y' \in \mathcal{Y}} [\mathbf{L}]_{yy'}\left((1 - [\mathbf{C}]_{y'}\delta_{yy'} + [\mathbf{C}]_{y'}(1 - \delta_{yy'}))\right),$$

as well as a possible base loss (such as cross entropy) and regularizer. The $\log$ term in (3) is used for numerical stability reasons for the gradient-based optimizers employed by the authors. Let us abstract these factors into the problem

$$\min_{\theta \in \Theta} \{L(\theta) := h(\mathbb{E}\left[\ell(\theta, \tau(\theta), X, Y)\right]) + R(\theta)\}, \tag{ConfTr-risk}$$

to be solved via a gradient-based method for some monotone transformation $h(\cdot)$, conformal loss $\ell(\cdot)$, and regularizer $R(\cdot)$. The underlying assumption, just as in any supervised learning task, is that the marginal distribution of $(X, Y)$ is unknown but that instead we can collect some i.i.d. training data $\mathcal{D} = \{(X_1, Y_1), \ldots, (X_n, Y_n)\}$. With this, an issue presents itself in that, unlike a typical loss function, we cannot evaluate $\frac{\partial}{\partial \theta}[\ell(\theta, \tau(\theta), X_i, Y_i)]$ from knowledge alone of $\theta, X_i, Y_i$, because $\tau(\theta) = Q_\alpha(E_\theta(X, Y))$ is a function of the *distribution* of $(X, Y)$ and not a mere transformation. To resolve this issue, Stutz et al. (2022) propose their `ConfTr` algorithm, which randomly splits

a given batch $B$ into two parts, which they refer to as *calibration* batch $B_{\text{cal}}$ and *prediction* batch $B_{\text{pred}}$. With this, the authors advocate for employing any smooth (differentiable) quantile estimator algorithm for $\tau(\theta)$ using the calibration batch. Then, they propose using this estimator to compute a sampled approximation of (ConfTr-risk), replacing expectations by sample means constructed using the prediction batch. Let $\hat{L}(\theta)$ denote the end-to-end empirical approximation of $L(\theta)$ in terms of the current parameters $\theta$. Once $\hat{L}(\theta)$ is constructed, the authors advocate for a (naive) risk minimization procedure where $\frac{\partial \hat{L}}{\partial \theta}(\theta)$ is computed and passed to an optimizer of choice.

## 2.3 VARIANCE ANALYSIS FOR CONFTR

Consider i.i.d. samples $\{(X_i, Y_i)\}_{i=1}^n$ from which we seek to estimate $\tau(\theta) = Q_\alpha(E_\theta(X, Y))$ and let $E_{(1)}(\theta) \leq \ldots \leq E_{(n)}(\theta)$ denote the order statistics corresponding to the scalar random variables $E_\theta(X_1, Y_1), \ldots, E_\theta(X_n, Y_n)$. Unlike the expectation and covariance matrix of a random vector, there is no universal consensus on an estimator for the population quantile of scalar random variables. Hyndman & Fan (1996) summarized and unified a significant portion of the various estimators proposed in the literature at the time. Following the notation of the aforementioned work, we will consider estimators of the form

$$\hat{\tau}(\theta) = \gamma E_{(j)}(\theta) + (1 - \gamma) E_{(j+1)}(\theta) \tag{4}$$

for some $\gamma = \gamma(j, g) \in [0, 1]$ with $j = \lfloor \alpha n + r \rfloor$ and $g = \alpha n + r - j$, where $r \in \mathbb{R}$ is a hyper-parameter. Other estimators have been proposed since, and even at the time of (Hyndman & Fan, 1996). However, the majority of statistical packages implement, by default, an estimator of the form (4). Other approaches have been proposed in the literature, via kernel-based methods, variational methods, and dispersion-based methods. With this estimator of equation (4), we see that

$$\frac{\partial \hat{\tau}}{\partial \theta}(\theta) = \gamma \frac{\partial E_{(j)}}{\partial \theta}(\theta) + (1 - \gamma) \frac{\partial E_{(j+1)}}{\partial \theta}(\theta). \tag{5}$$

Note that $\{E_{(i)}(\theta)\}_{i=1}^n$ are differentiable almost surely (see Appendix B.1 for more details). Further, if $\omega(\theta) : [n] \to [n]$ denotes the permutation of indices $[n] := \{1, \ldots, n\}$ that correspond to the order statistics, i.e. $E_{(j)}(\theta) = E_\theta(X_{\omega_j(\theta)}, Y_{\omega_j(\theta)})$ with $\omega(\theta) = (\omega_1(\theta), \ldots, \omega_n(\theta))$, we see that $\omega(\theta)$ is piecewise constant (or approximately so when using a smooth sorting such as in (Blondel et al., 2020; Cuturi et al., 2019)), and thus $\frac{\partial \omega}{\partial \theta}(\theta) \approx 0$. By the chain rule, it follows that $\frac{\partial E_{(j)}}{\partial \theta}(\theta) \approx \frac{\partial E}{\partial \theta}(\theta, X_{\omega_j(\theta)}, Y_{\omega_j(\theta)})$, where $E(\theta, X, Y) = E_\theta(X, Y)$. Since $E_{(j)}(\theta) \approx \tau(\theta)$ and $E_{(j+1)}(\theta) \approx \tau(\theta)$, and noting that the samples $(X_1, Y_1), \ldots, (X_n, Y_n)$ are i.i.d., then $(X_{\omega_j(\theta)}, Y_{\omega_j(\theta)})$ and $(X_{\omega_{j+1}(\theta)}, Y_{\omega_{j+1}(\theta)})$ are approximately independent and approximately distributed as equal to the distribution of $(X, Y)$ when conditioned on $E_\theta(X, Y) = \tau(\theta)$. Hence,

$$\mathbb{E}\left[\frac{\partial \hat{\tau}}{\partial \theta}(\theta)\right] \approx \mathbb{E}\left[\frac{\partial E}{\partial \theta}(\theta, X, Y) \,\Big|\, E_\theta(X, Y) = \tau(\theta)\right] \tag{6}$$

$$\text{cov}\left(\frac{\partial \hat{\tau}}{\partial \theta}(\theta)\right) \approx (\gamma^2 + (1 - \gamma)^2) \text{cov}\left(\frac{\partial E}{\partial \theta}(\theta, X, Y) \,\Big|\, E_\theta(X, Y) = \tau(\theta)\right). \tag{7}$$

Inspecting (7), we can see that the variance of the naive estimator $\frac{\partial \hat{\tau}}{\partial \theta}(\theta)$ for $\frac{\partial \tau}{\partial \theta}(\theta)$ is approximately constant when the sample size is moderately large. In particular, the variance is approximately $\mathcal{O}(1)$, which is quite sample inefficient as it does not decrease as the sample size increases. On the other hand, by differentiating the conformal training loss, i.e.

$$\frac{\partial}{\partial \theta}[\ell(\theta, \hat{\tau}(\theta), x, y)] = \frac{\partial \ell}{\partial \theta}(\theta, \hat{\tau}(\theta), x, y) + \frac{\partial \ell}{\partial \tau}(\theta, \hat{\tau}(\theta), x, y)\frac{\partial \hat{\tau}}{\partial \theta}(\theta), \tag{8}$$

it becomes apparent that poor estimator variance for $\frac{\partial \hat{\tau}}{\partial \theta}(\theta)$ will bottleneck sample efficiency in the estimation of $\frac{\partial L}{\partial \theta}(\theta)$ obtained by replacing $\tau(\theta)$ in (ConfTr-risk) with $\hat{\tau}(\theta)$ and using the prediction batch to approximate the expectations. Note that (8) follows from the chain rule, see Appendix B.2 for more details. In the next section, we present our proposed solution to address this issue.

## 3 VARIANCE-REDUCED CONFORMAL TRAINING

In order to surpass the shortcoming of `ConfTr` described in the previous section, let us first note that the gradient of the conformal risk (ConfTr-risk) can be written as

$$\frac{\partial L}{\partial \theta}(\theta) = h'(\mathbb{E}\left[\ell(\theta, \tau(\theta), Z)\right]) \left( \mathbb{E}\left[\frac{\partial \ell}{\partial \theta}(\theta, \tau(\theta), Z)\right] + \mathbb{E}\left[\frac{\partial \ell}{\partial \tau}(\theta, \tau(\theta), Z)\right] \frac{\partial \tau}{\partial \theta}(\theta) \right), \quad (9)$$

where $h'$ denotes the derivative of $h$, $Z = (X, Y)$, and noting that we dropped the regularizer for simplicity. Additionally, we can exploit the following relationship to further characterize `ConfTr` as well as to design a variance-reduced alternative:

**Proposition 3.1** (Quantile Sensitivity (Hong, 2009)). *Suppose that $X$ is absolutely continuous and $E_\theta(x, y)$ is continuously differentiable in $\theta$ and $x$. Then, for every $\theta \in \Theta$,*

$$\frac{\partial \tau}{\partial \theta}(\theta) = \mathbb{E}\left[\frac{\partial E}{\partial \theta}(\theta, X, Y) \,\Big|\, E_\theta(X, Y) = \tau(\theta)\right]. \quad (10)$$

In Appendix A, we provide a rigorous proof for the above proposition, which was carried out independently from that of the equivalent result of Hong (2009) (namely, Theorem 2). However, note that the assumptions in (Hong, 2009) are less restrictive than the ones we use. Further, the author explores more deeply the connections between $\tau(\theta)$ and $\frac{\partial \tau}{\partial \theta}(\theta)$.

Equipped with the above proposition, we can compare (6) and (10) to see that, despite the poor sample efficiency of the naive estimator for $\frac{\partial \tau}{\partial \theta}(\theta)$, it at least leads to an approximately unbiased estimator. However, it also becomes intuitively clear that variance reduction can be achieved by exploiting (10), for example by decoupling the estimation of $\tau(\theta)$ from $\frac{\partial \tau}{\partial \theta}(\theta)$, and generally by not settling for $\widehat{\frac{\partial \tau}{\partial \theta}}(\theta) := \frac{\partial \hat{\tau}}{\partial \theta}(\theta)$ as the estimator for $\frac{\partial \tau}{\partial \theta}(\theta)$.

### 3.1 QUANTILE GRADIENT ESTIMATION

We can use Proposition 3.1 to design an algorithm that boosts the estimated quantile gradient. The idea is as follows: if we denote

$$\eta(\theta) := \mathbb{E}\left[\frac{\partial E}{\partial \theta}(\theta, X, Y) \,\big|\, A(\theta)\right], \qquad \Sigma(\theta) := \text{cov}\left(\frac{\partial E}{\partial \theta}(\theta, X, Y) \,\big|\, A(\theta)\right), \quad (11)$$

$$\eta_\varepsilon(\theta) := \mathbb{E}\left[\frac{\partial E}{\partial \theta}(\theta, X, Y) \,\big|\, A_\varepsilon(\theta)\right], \qquad \Sigma_\varepsilon(\theta) := \text{cov}\left(\frac{\partial E}{\partial \theta}(\theta, X, Y) \,\big|\, A_\varepsilon(\theta)\right), \quad (12)$$

for $\varepsilon > 0$, where $A(\theta) := \{E_\theta(X, Y) = \tau(\theta)\}$ and $A_\varepsilon(\theta) := \{|E_\theta(X, Y) - \tau(\theta)| \leq \varepsilon\}$, then the terms in (11) and (12) are approximately equal if $\varepsilon \approx 0$. Subsequently, we can more efficiently estimate $\eta(\theta) = \frac{\partial \tau}{\partial \theta}(\theta)$, compared to the naive estimator $\hat{\eta}(\theta) = \frac{\partial \hat{\tau}}{\partial \theta}(\theta)$, by naive sample estimate of $\eta_\varepsilon(\theta)$. To this end, we propose the following $\varepsilon$-estimator

$$\hat{\eta}(\theta) := \frac{1}{\sum_{i=1}^n 1_{\hat{A}_{\varepsilon,i}(\theta)}} \sum_{i=1}^n 1_{\hat{A}_{\varepsilon,i}(\theta)} \frac{\partial E}{\partial \theta}(\theta, X_i, Y_i), \quad (13)$$

from i.i.d. copies $(X_1, Y_1), \ldots, (X_n, Y_n)$ of $(X, Y)$, where $\hat{A}_{\varepsilon,i}(\theta) = \{|E_\theta(X_i, Y_i) - \hat{\tau}(\theta)| \leq \varepsilon\}$.

Alternative estimators for $\eta(\theta)$ can be constructed. Some examples include:

- **Ranking**: sort the examples based on the distances $\{|E_\theta(X_i, Y_i) - \hat{\tau}(\theta)|\}_{i=1,\ldots,n}$, choose the "top" $m$ samples (smallest distances) for some suitably small $m$, and then set $\hat{\eta}(\theta)$ as the average of $\frac{\partial E}{\partial \theta}(\theta, X_i, Y_i)$ over those samples. Note that this can be seen as an heuristic to choose $\varepsilon$ when using the $\varepsilon$-estimator, and it is the strategy we adopt in our experiments.
- **Kernel regression**: consider some kernel $K_h(t) = \frac{1}{h}K\left(\frac{t}{h}\right)$ with $h > 0$ for which $\int_{-\infty}^{+\infty} tK(t)\mathrm{d}t = 0$, for instance $K(\cdot) = $ PDF of $\mathcal{N}(0, 1)$ or $K(t) = \sigma(t)(1 - \sigma(t))$ where $\sigma(\cdot)$ denotes the sigmoid function. Then, we can use the (modified) Nadaraya–Watson estimator $\hat{\eta}(\theta) = \frac{\sum_{i=1}^n K_h(E_\theta(X_i, Y_i) - \hat{\tau}(\theta)) \frac{\partial E}{\partial \theta}(\theta, X_i, Y_i)}{\sum_{i=1}^n K_h(E_\theta(X_i, Y_i) - \hat{\tau}(\theta))}$.

- **Random splitting**: split the $n$ examples into $m$ sub-sets of samples, apply some other estimator algorithm for $\eta(\theta)$ on each sub-set, and then set $\hat{\eta}(\theta)$ as the average of the individual estimates of $\eta(\theta)$.

Other closely related notions that could be adapted for the estimator $\hat{\eta}(\theta)$ and that could lead to reduced variance include importance sampling and smooth bootstrapping, but more generally, averages can be replaced by carefully constructed weighted sums. Particle filters could be feasible as a way of more efficiently estimate $\eta(\theta)$ by carrying over the previous estimates as $\theta$ is updated. It is also clear that various of these potential estimators are closely linked. For instance, if the threshold $\varepsilon > 0$ is allowed to be sample-dependent, then the thresholding approach (13) and the ranking approach can be seen as equivalent by setting $\varepsilon = \inf\left\{ \varepsilon' > 0 : \sum_{i=1}^{n} 1_{\hat{A}_{\varepsilon',i}(\theta)} \geq m \right\}$.

### 3.2 PROPOSED ALGORITHM: `VR-CONFTR`

Suppose that a variance-reduced estimator for $\frac{\partial \tau}{\partial \theta}(\theta)$ has been already designed. Then, the new estimate for $\tau(\theta)$ and $\frac{\partial \tau}{\partial \theta}(\theta)$ can be plugged into expression (9) for the gradient of the conformal training risk function, before the expectations can be approximated by sample means, leading to the *plug-in* estimator for $\frac{\partial L}{\partial \theta}(\theta)$. Naturally, the plug-in gradient estimator is then passed through an optimizer in order to approximately solve (CRM). Our proposed pipeline, which we call *variance-reduced conformal training* (`VR-ConfTr`) algorithm, constitutes our main contribution and proposed solution to improve the sample inefficiency of `ConfTr`. The critical step of constructing the plug-in estimator is summarized in Algorithm 1. Additionally, the entire pipeline is illustrated in Figure 1.

---

**Algorithm 1** Variance-reduced conformal training (`VR-ConfTr`)

---

**Require:**

    batch $B = \{(X_1, Y_1), \ldots, (X_{2n}, Y_{2n})\}$ of i.i.d. samples from $(X, Y)$,

    score function $E(\theta, x, y) : \Theta \times \mathcal{X} \times \mathcal{Y} \to \mathbb{R}$,

    conformal loss $\ell(\theta, x, y, \tau) : \Theta \times \mathcal{X} \times \mathcal{Y} \times \mathbb{R} \to \mathbb{R}$,

    monotone transformation $\mathcal{F} : \mathbb{R} \to \mathbb{R}$,

    estimator $\hat{\tau}(\cdot)$ for $\tau(\theta) = Q_\alpha(E_\theta(X, Y))$,

    estimator $\widehat{\frac{\partial \tau}{\partial \theta}}(\cdot)$ for $\frac{\partial \tau}{\partial \theta}(\theta)$.

**Ensure:** output an estimate $\widehat{\frac{\partial L}{\partial \theta}}$ of the gradient $\frac{\partial L}{\partial \theta}(\theta)$ of the conformal training risk (ConfTr-risk)

  1: partition $B$ into $\{B_{\text{cal}}, B_{\text{pred}}\}$, with $|B_{\text{cal}}| = |B_{\text{pred}}| = n$.

  2: $\hat{\tau} \leftarrow \hat{\tau}(B_{\text{cal}})$                                            // estimate $\tau(\theta)$ using $B_{\text{cal}}$

  3: $\widehat{\frac{\partial \tau}{\partial \theta}} \leftarrow \widehat{\frac{\partial \tau}{\partial \theta}}(B_{\text{cal}})$                            // estimate $\frac{\partial \tau}{\partial \theta}(\theta)$ using $B_{\text{cal}}$

  4: $\hat{\ell} \leftarrow \frac{1}{|B_{\text{pred}}|} \sum_{(x,y) \in B_{\text{pred}}} \ell(\theta, x, y, \hat{\tau})$

  5: $\widehat{\frac{\partial \ell}{\partial \theta}} \leftarrow \frac{1}{|B_{\text{pred}}|} \sum_{(x,y) \in B_{\text{pred}}} \frac{\partial \ell}{\partial \theta}(\theta, x, y, \hat{\tau})$

  6: $\widehat{\frac{\partial \ell}{\partial \tau}} \leftarrow \frac{1}{|B_{\text{pred}}|} \sum_{(x,y) \in B_{\text{pred}}} \frac{\partial \ell}{\partial \tau}(\theta, x, y, \hat{\tau})$

  7: $\widehat{\frac{\partial L}{\partial \theta}} \leftarrow h'(\hat{\ell}) \left( \widehat{\frac{\partial \ell}{\partial \theta}} + \widehat{\frac{\partial \ell}{\partial \tau}} \widehat{\frac{\partial \tau}{\partial \theta}} \right) + \frac{\partial R}{\partial \theta}(\theta)$          // **"plug-in"** gradient estimator

  8: **return** $\widehat{\frac{\partial L}{\partial \theta}}$

---

### 3.3 THEORETICAL RESULTS

We focus our theoretical analysis on the thresholding esimator (13). Note that the $m$-ranking estimator, which we use in our experiments, is effectively an $\varepsilon$-threshold estimator where the ranking is a heuristic criterion to choose the threshold $\varepsilon$ at each iteration based on the current batch and parameter. For simplicity and to avoid having to commit to any particular quantile estimator, we assume $\hat{\tau}(\theta) = \tau(\theta)$ in the analysis. Furthermore, we will assume $\varepsilon > 0$ to be deterministic. Lastly, we will assume that in the event in which $\bigcup_{i=1}^{n} \hat{A}_{\varepsilon,i}(\theta)$ is empty, the estimator evaluates to $\hat{\eta}(\theta) = 0$. With this, we can establish our main theoretical result in the following theorem.

**Theorem 3.1** (Variance reduction). *Let $\hat{\eta}(\theta)$ be the estimator defined in (13) with $\hat{\tau}(\theta) = \tau(\theta)$. Then, the the bias and variance of the estimator can be characterized as follows:*

$$(i) \qquad \mathbb{E}\left[\hat{\eta}(\theta)\right] = (1 - [q_\varepsilon(\theta)]^n)\eta_\varepsilon(\theta) \qquad \text{(bias)}$$

$$(ii) \qquad \text{cov}\left(\hat{\eta}(\theta)\right) \preceq \frac{2\Sigma_\varepsilon(\theta)}{p_\varepsilon(\theta)n} + [q_\varepsilon(\theta)]^n \eta_\varepsilon(\theta)\eta_\varepsilon^\mathsf{T}(\theta), \qquad \text{(variance)}$$

*where $p_\varepsilon(\theta) = \mathbb{P}(A_{\varepsilon,i}(\theta))$ and $q_\varepsilon(\theta) = 1 - p_\varepsilon(\theta)$.*

The main takeaway of result $(i)$ is that $\hat{\eta}(\theta)$ is an *asymptotically unbiased* estimator of $\eta_\varepsilon(\theta)$, but not $\eta(\theta)$. However, by definition we also have $\eta_\varepsilon(\theta) \approx \eta(\theta)$ for $\varepsilon \approx 0$. The second result $(ii)$, instead, shows that *variance reduction* is obtained by the proposed estimator, when compared to the naive estimator $\frac{\partial \hat{\tau}}{\partial \theta}(\theta)$. Further, for large $n$, the variance reduction is proportional to $p_\varepsilon(\theta)n$, which is equal to the (expected) proportion of samples that are ultimately used in the estimator. More precisely, the variance of the estimator is $\mathcal{O}\left(\frac{1}{p_\varepsilon(\theta)n}\right)$ as $\varepsilon \to 0$ or $n \to \infty$.

A key takeaway of $(i)$ and $(ii)$ is the explicit characterization of the *bias-variance trade-off* as a function of the threshold $\varepsilon > 0$ and of the batch size $n$: for a given batch size $n$, a larger $\varepsilon$ increases the expected amount of samples used by the estimator, thus reducing its variance. However, larger $\varepsilon$ also increases the bias of the estimator towards the unconditional expectation $\mathbb{E}\left[\frac{\partial E}{\partial \theta}(\theta, X, Y)\right]$, where we make note that $\eta_\varepsilon(\theta) \to \eta(\theta)$ as $\varepsilon \to 0$.

## 4 EXPERIMENTS

As a warm-up, we illustrate Theorem 3.1 on a synthetic Gaussian mixture model (GMM) dataset, depicted in Figure 2. We employ the $m$-ranking method with top $m = \frac{\alpha n}{\log \log n}$ samples. This ratio performs well across a variety of settings. As shown, our estimator (`VR-ConfTr`) reduces variance effectively, while the naive one (`ConfTr`) is sample inefficient.

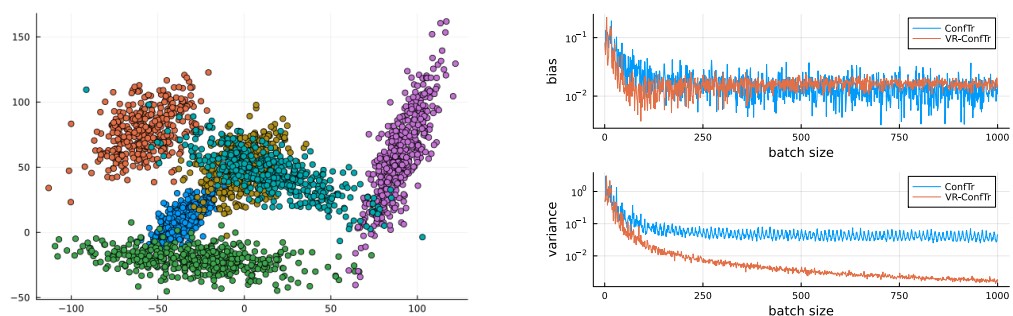

Figure 2: Sample batch from GMM distribution (left) and corresponding bias and variance for the quantile gradient estimates (right).

### 4.1 BENCHMARK DATASETS AND ALGORITHMS

We evaluate the effectiveness of `VR-ConfTr` against (i) a baseline model trained with standard cross-entropy loss (we refer to this method simply as `Baseline`), and (ii) the `ConfTr` algorithm proposed by Stutz et al. (2022). We perform experiments across benchmark datasets - MNIST Deng (2012), Fashion-MNIST Xiao et al. (2017a), Kuzushiji-MNIST Clanuwat et al. (2018) -, and a healthcare dataset comprising abdominal computed tomography scans, OrganAMNIST Yang et al. (2021). One of the main performance metrics that we consider is the *length-efficiency* of the conformal prediction sets produced by applying a standard CP procedure to the trained model. Other relevant metrics are the convergence speed and the variance across multiple runs. For the choice of the quantile gradient estimator $\widehat{\frac{\partial \tau}{\partial \theta}}$ of `VR-ConfTr`, we use the $m$-ranking approach presented in section 3.1. We choose this estimator because it is the more closely related to the one analyzed in Theorem 3.1. We investigate multiple possibilities for the choice of $m$, and more details on this

tuning can be found in Appendix C. We provide extensive details about the training settings, the adopted model architectures, hyper-parameters and additional results in Appendix D.

In the next subsection, we present the summary of results obtained from evaluating the model *after training*. Given a number of epochs, we train a model over multiple runs. For the obtained model, we show: (i) the average accuracy and its standard deviation, and (ii), the average length efficiency and its standard deviation. In section 4.3, to illustrate further the improved training performance of `VR-ConfTr` over the original `ConfTr` algorithm and the variance reduction effect, we show the trajectories of relevant evaluation metrics - the conformal training loss defined in section 2, and the length efficiency - for all datasets and methods during training.

## 4.2 SUMMARY OF EVALUATION RESULTS

Table 1 presents the inefficiency results of the CP procedure applied post-training, and the accuracy of the trained model for each dataset, with the corresponding standard deviations.

| Dataset | Algorithm | Accuracy (Avg ± Std) | Avg Size | Std Size |
|---|---|---|---|---|
| MNIST | Baseline | $0.887 \pm 0.004$ | 4.122 (+12%) | 0.127 |
| | ConfTr Stutz et al. (2022) | $0.842 \pm 0.141$ | 3.990 (+8%) | 0.730 |
| | VR-ConfTr (**ours**) | $0.886 \pm 0.071$ | **3.688** | 0.350 |
| Fashion-MNIST | Baseline | $0.845 \pm 0.002$ | 3.218 (+15%) | 0.048 |
| | ConfTr Stutz et al. (2022) | $0.799 \pm 0.065$ | 3.048 (+9%) | 0.201 |
| | VR-ConfTr (**ours**) | $0.839 \pm 0.043$ | **2.795** | 0.154 |
| Kuzushiji-MNIST | Baseline | $0.872 \pm 0.046$ | 4.982 (+6%) | 0.530 |
| | ConfTr Stutz et al. (2022) | $0.783 \pm 0.125$ | 4.762 (+2%) | 0.226 |
| | VR-ConfTr (**ours**) | $0.835 \pm 0.098$ | **4.657** | 0.680 |
| OrganA-MNIST | Baseline | $0.552 \pm 0.017$ | 4.823 (+2%) | 0.748 |
| | ConfTr Stutz et al. (2022) | $0.526 \pm 0.047$ | 6.362 (+33%) | 0.857 |
| | VR-ConfTr (**ours**) | $0.547 \pm 0.021$ | **4.776** | 1.178 |

Table 1: Summary of evaluation results. For `VR-ConfTr`, we show in percentage the average set size (**Avg Size**) improvement against `ConfTr` by Stutz et al. (2022). The third column presents the average accuracy and its standard deviation (**Accuracy (Avg ± Std)**).

The metrics reported in Table 1 are computed as averages over 5-10 training trials depending on the dataset. The way in which the number of the random training trials varies across the datasets is discussed in more detail in appendix D. Similarly to the approach followed by Stutz et al. (2022), we are mostly interested in the effectiveness of the different algorithms on the CP efficiency, and therefore we do not focus on improving the accuracy by using more advanced model architectures. To ensure a fair comparison for each dataset, we used the same exact model architecture across the three different methods (`ConfTr`, `VR-ConfTr` and `Baseline`). Furthermore, the training and evaluation hyper-parameters are identical across `ConfTr` and `VR-ConfTr`. For the CP procedure applied post-training, we use the standard `THR` method with $\alpha = 0.01$. The average set-size for each method is reported over 10 different splits of the calibration and test data used for the conformal prediction procedure. The main takeaway from Table 1 is that `VR-ConfTr` improves over all considered metrics compared to `ConfTr`.

In terms of "length-efficiency", `VR-ConfTr` is able to consistently achieve smaller prediction set sizes compared to both `ConfTr` and `Baseline`. It is important to note that the focus of our work is not to tune `ConfTr` to achieve better performance than `Baseline`, but rather to show that regardless of the performance of `ConfTr` and the hyper-parameters chosen, `VR-ConfTr` effectively provide performance improvements and training stability with the same hyper-parameters. Note that, similar to the results reported by Stutz et al. (2022), the `Baseline` architecture is sometimes able to achieve slightly higher accuracy than `ConfTr` and `VR-ConfTr`. It can be seen that `VR-ConfTr` consistently achieves higher accuracy compared to `ConfTr`. However, we stress that the objective of conformal training is to reduce the size of the prediction sets while preserving a similar accuracy as non-conformal training, and not to improve the accuracy.

## 4.3 ON THE TRAINING PERFORMANCE OF VR-ConfTr

Here, we focus on the training performance of VR-ConfTr, with special attention to the speed in minimizing the conformal training loss described in section 2, and in minimizing the CP set sizes on test data. The results, which we illustrate plotting the evolution of the different metrics across epochs, validate the beneficial effect of the variance reduction technique and the superior performance of VR-ConfTr when compared to the competing ConfTr by Stutz et al. (2022).

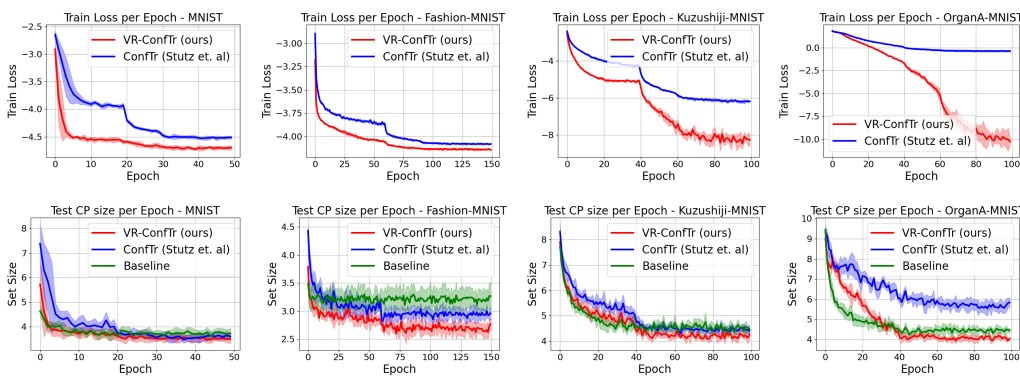

Figure 3: Training curves for MNIST, Fashion-MNIST, Kuzushiji-MNIST, and OrganAMNIST. For each dataset, we show the training loss on top and corresponding test CP set sizes at the bottom at the end of each epoch, evaluated using the THR conformal predictor.

In Figure 3, we show the training performance for four datasets (MNIST, FMNIST, KMNIST and OrganAMNIST) illustrating two key metrics: (i) the evolution of the conformal training loss defined in section 2 and (ii) the test CP size across epochs. In the four plots on top, we show the comparison between the train loss evolution obtained using our VR-ConfTr against the one obtained by ConfTr. In the four plots at the bottom, we show the comparison between the test CP set sizes for VR-ConfTr, ConfTr and Baseline. In all the plots, we see that VR-ConfTr reaches smaller values of the loss and in significantly fewer epochs as compared to ConfTr. In the case of MNIST, for example, VR-ConfTr reaches a lower value of the loss in 10 times fewer epochs as compared to ConfTr. Similarly, for FMNIST VR-ConfTr achieves a smaller size in one third of epochs compared to ConfTr. For both Kuzushiji-MNIST and OrganA-MNIST, we notice that not only VR-ConfTr is faster, but it also gets to significantly smaller values of the loss. For the more challenging OrganA-MNIST dataset, this difference appears even more accentuated, not only in the training loss but also in the test CP set sizes. Notice that for all the three methods (VR-ConfTr, ConfTr and Baseline) we performed hyper-parameters tuning. Notably, in the case of the OrganA-MNIST dataset, we were not able to obtain an improvement with ConfTr in the final set size with respect to Baseline, which stresses the need for a method with improved gradient estimation, as the one we propose in this paper. More details on the grid-search over hyper-parameters and additional experiments for all algorithms can be found in appendix C and D.

## 5 CONCLUDING REMARKS AND FUTURE DIRECTIONS

We formalized the concept of optimizing CP efficiency during training as the problem of *conformal risk minimization* (CRM). We identified a key source of sample inefficiency in the ConfTr method proposed by Stutz et al. (2022), which is a CRM method for length efficiency optimization. Our theoretical analysis elucidated the source of sample inefficiency, which lies in the estimation of the gradient of the population quantile. To address this issue, we introduced a novel technique that improves the gradient estimation of the population quantile of the conformity scores by provably reducing its variance. We show that, by incorporating this estimation technique in our proposed VR-ConfTr algorithm, the training becomes more stable and the post-training conformal predictor is often more efficient as well. Our work also opens up possibilities for future research in the area of CRM. Indeed, further methods for quantile gradient estimation could be developed and readily integrated with our "plug-in" algorithm, for which we can expect improved training performance.

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

# A  PROOFS

In this appendix, we provide the proofs of all the theoretical results presented in the paper.

## A.1  PROOF OF LEMMA 3.1

Let $H(s) = \begin{cases} 1, & s \geq 0 \\ 0, & s < 0 \end{cases}$ denote the Heaviside step function, and let $H_n(s) = \Phi\left(\frac{s}{\sigma_n}\right)$ denote a smooth approximation, where $\Phi$ denotes the cumulative distribution function (CDF) of the standard Gaussian distribution, and $\sigma_n > 0$ is a sequence such that $\sigma_n \to 0$ as $n \to \infty$. Note that $H_n(s) \to H(s)$ pointwise as $n \to \infty$, and that each $H_n$ is smooth.

By definition, we have

$$\mathbb{P}\left[E(\theta, X, Y) \leq \tau(\theta)\right] = \alpha. \tag{14}$$

which we can rewrite as

$$\mathbb{E}\left[H\big(\tau(\theta) - E(\theta, X, Y)\big)\right] = \alpha. \tag{15}$$

Since $0 \leq H(s) \leq 1$ for all $s$, and $H_n(s) \to H(s)$ pointwise, by the Dominated Convergence Theorem, we have

$$\begin{aligned}
\alpha &= \mathbb{E}\left[H\big(\tau(\theta) - E(\theta, X, Y)\big)\right] \\
&= \mathbb{E}\left[\lim_{n\to\infty} H_n\big(\tau(\theta) - E(\theta, X, Y)\big)\right] \\
&= \lim_{n\to\infty} \mathbb{E}\left[H_n\big(\tau(\theta) - E(\theta, X, Y)\big)\right].
\end{aligned} \tag{16}$$

Differentiating both sides with respect to $\theta$, we obtain

$$0 = \frac{\partial}{\partial\theta} \lim_{n\to\infty} \mathbb{E}\left[H_n\left(\tau(\theta) - E(\theta, X, Y)\right)\right], \tag{17}$$

where $\alpha$ is a constant independent of $\theta$. By interchanging the limit and differentiation, this becomes

$$0 = \lim_{n\to\infty} \frac{\partial}{\partial\theta} \mathbb{E}\left[H_n\left(\tau(\theta) - E(\theta, X, Y)\right)\right]. \tag{18}$$

To justify the interchange, we note that $f_n(\theta) = \mathbb{E}\left[H_n\left(\tau(\theta) - E(\theta, X, Y)\right)\right]$ converges pointwise to $\alpha$, a constant. By uniform convergence of $f_n(\theta)$ and its derivative $\frac{\partial}{\partial\theta}$, we can exchange the limit and differentiation. Using the Leibniz Integral Rule, we interchange differentiation and expectation:

$$0 = \lim_{n\to\infty} \mathbb{E}\left[\frac{\partial}{\partial\theta} H_n\left(\tau(\theta) - E(\theta, X, Y)\right)\right]. \tag{19}$$

The interchange is valid because $H_n$ is infinitely differentiable, $\tau(\theta)$ and $E(\theta, X, Y)$ are continuously differentibale with respect to $\theta$, and the derivative $\frac{\partial}{\partial\theta} H_n(\tau(\theta) - E(\theta, X, Y))$ is continuous in $\theta$ and integrable. Finally applying the chain rule to differentiate $H_n\left(\tau(\theta) - E(\theta, X, Y)\right)$ with respect to $\theta$, we get:

$$0 = \lim_{n\to\infty} \mathbb{E}\left[H_n'\left(\tau(\theta) - E(\theta, X, Y)\right)\left(\frac{\partial\tau}{\partial\theta}(\theta) - \frac{\partial E}{\partial\theta}(\theta, X, Y)\right)\right]. \tag{20}$$

Define

$$\delta_n(s) = H_n'(s) = \frac{1}{\sqrt{2\pi}\sigma_n} e^{-\frac{s^2}{2\sigma_n^2}},$$

$$\gamma_n(\theta, x, y) = \delta_n\left(\tau(\theta) - E(\theta, x, y)\right),$$

$$\Delta(\theta, x, y) = \frac{\partial\tau}{\partial\theta}(\theta) - \frac{\partial E}{\partial\theta}(\theta, x, y).$$

Now, we can see that

$$\frac{\partial}{\partial\theta} \mathbb{E}\left[H_n\left(\tau(\theta) - E(\theta, X, Y)\right)\right] = \mathbb{E}\left[\gamma_n(\theta, X, Y)\Delta(\theta, X, Y)\right]. \tag{21}$$

Let $a_n \asymp b_n$ denote asymptotic equivalence, meaning that $\lim_{n \to \infty} \frac{a_n}{b_n} = 1$, assuming $b_n \neq 0$ for finite $n$. From equation (20), it follows that

$$\mathbb{E}\left[\gamma_n(\theta, X, Y)\Delta(\theta, X, Y)\right] \asymp 0. \tag{22}$$

**Analyzing the Expectation:**

Let $\varepsilon_n > 0$ be any sequence such that $\varepsilon_n = o(\sigma_n)$, meaning $\varepsilon_n/\sigma_n \to 0$ as $n \to \infty$. Define the set

$$A_{\varepsilon_n}(\theta) = \left\{\omega \in \Omega : -\varepsilon_n < E(\theta, X(\omega), Y(\omega)) - \tau(\theta) < \varepsilon_n\right\}. \tag{23}$$

We can decompose the expectation in (22) as

$$\mathbb{E}\left[\gamma_n\Delta\right] = \mathbb{P}\left(A_{\varepsilon_n}(\theta)\right)\mathbb{E}\left[\gamma_n\Delta \mid A_{\varepsilon_n}(\theta)\right] + \mathbb{P}\left(A_{\varepsilon_n}^c(\theta)\right)\mathbb{E}\left[\gamma_n\Delta \mid A_{\varepsilon_n}^c(\theta)\right]. \tag{24}$$

**Negligibility of the Second Term**: On the complement $A_{\varepsilon_n}^c(\theta)$, the value $\tau(\theta) - E(\theta, X, Y)$ is either greater than $\varepsilon_n$ or less than $-\varepsilon_n$. Therefore, for $s \geq \varepsilon_n$ or $s \leq -\varepsilon_n$, $\delta_n(s) = H_n'(s)$ becomes very small. Particularly

$$\delta_n(s) = \frac{1}{\sqrt{2\pi}\,\sigma_n}\exp\left(-\frac{s^2}{2\sigma_n^2}\right).$$

Since $\varepsilon_n = o(\sigma_n)$ and $\sigma_n \to 0$, it follows that $\frac{\varepsilon_n}{\sigma_n} \to \infty$. For $|s| \geq \varepsilon_n$, we have: $\delta_n(s) \leq$

$\frac{1}{\sqrt{2\pi}\,\sigma_n}\exp\left(-\frac{\varepsilon_n^2}{2\sigma_n^2}\right)$. from $\frac{\varepsilon_n}{\sigma_n} \to \infty$, it also follows that $\exp\left(-\frac{\varepsilon_n^2}{2\sigma_n^2}\right) \to 0$ and

$$\gamma_n(\theta, X, Y) \leq \frac{1}{\sqrt{2\pi}\,\sigma_n}\exp\left(-\frac{\varepsilon_n^2}{2\sigma_n^2}\right) \to 0.$$

Therefore, (24) becomes

$$\mathbb{E}\left[\gamma_n\Delta\right] \asymp \mathbb{P}\left(A_{\varepsilon_n}(\theta)\right)\mathbb{E}\left[\gamma_n\Delta \mid A_{\varepsilon_n}(\theta)\right]. \tag{25}$$

Plugging in (25) into (22) we get:

$$\mathbb{E}[\gamma_n(\theta, X, Y)\Delta(\theta, X, Y) \mid A_{\varepsilon_n}(\theta)] \asymp 0,$$

By noting that $\mathbb{P}(A_{\varepsilon_n}(\theta)) > 0$ for all $n$ due to continuity of $x \mapsto E(\theta, x, y)$. We can then rewrite as

$$\mathbb{E}[\gamma_n(\theta, X, Y), \mid A_{\varepsilon_n}(\theta)]\frac{\partial \tau}{\partial \theta}(\theta) \asymp \mathbb{E}\left[\gamma_n(\theta, X, Y)\frac{\partial E}{\partial \theta}(\theta, X, Y) \mid A_{\varepsilon_n}(\theta)\right]. \tag{26}$$

When conditioned on $A_{\varepsilon_n}(\theta)$, we have $\gamma_n(\theta, X, Y) \asymp \delta_n(0)$. Indeed, note that $\delta_n(\varepsilon_n) \leq \gamma_n(\theta, X, Y) \leq \delta_n(0)$, which we can rewrite as $\frac{\delta_n(\varepsilon_n)}{\delta_n(0)} \leq \frac{\gamma_n(\theta, X, Y)}{\delta_n(0)} \leq 1$. Noting that $\delta_n(\varepsilon_n) \asymp 1$, it indeed follows that $\gamma_n(\theta, X, Y) \asymp \delta_n(0)$. Therefore, (26) can be further simplified to

$$\frac{\partial \tau}{\partial \theta}(\theta) \asymp \mathbb{E}\left[\frac{\partial E}{\partial \theta}(\theta, X, Y) \mid A_{\varepsilon_n}(\theta)\right], \tag{27}$$

which readily leads to

$$\frac{\partial \tau}{\partial \theta}(\theta) = \lim_{n \to \infty}\mathbb{E}\left[\frac{\partial E}{\partial \theta}(\theta, X, Y) \mid A_{\varepsilon_n}(\theta)\right]$$

$$= \lim_{\varepsilon \to 0}\mathbb{E}\left[\frac{\partial E}{\partial \theta}(\theta, X, Y) \mid A_\varepsilon(\theta)\right]$$

$$= \lim_{\varepsilon \to 0}\mathbb{E}\left[\frac{\partial E}{\partial \theta}(\theta, X, Y) \mid -\varepsilon < E_\theta(X, Y) - \tau(\theta) < \varepsilon\right]$$

$$= \mathbb{E}\left[\frac{\partial E}{\partial \theta}(\theta, X, Y) \mid E_\theta(X, Y) = \tau(\theta)\right].$$

## A.2 PROOF OF THEOREM 3.1

We start with two preliminary results that we will use in the proof.

**Some preliminaries.** First, we recall a well-known result. Let $k \sim \text{Binomial}(n, p)$ be a random variable sampled from a Binomial distribution with $n$ trials and with probability $p$ of success. The following holds:

$$\mathbb{E}\left[\frac{1}{1+k}\right] = \frac{(1 - (1-p)^{n+1})}{(n+1)p}. \tag{28}$$

Note that this follows from the following simple steps:

$$\begin{aligned}
\mathbb{E}\left[\frac{1}{1+k}\right] &= \sum_{k=0}^{n} \frac{1}{1+k} \cdot \binom{n}{k} p^k (1-p)^{n-k} \\
&= \frac{1}{p(n+1)} \sum_{k=0}^{n} \binom{n+1}{k+1} p^{k+1} (1-p)^{n-k} \\
&= \frac{1}{p(n+1)} \sum_{j=1}^{n+1} \binom{n+1}{j} p^j (1-p)^{n+1-j} \\
&= \frac{\left(1 - (1-p)^{n+1}\right)}{p(n+1)}.
\end{aligned}$$

Next, we state another well-known identity. Let us consider the following recursion:

$$a_{n+1} = \rho\, a_n + b,$$

where $\rho > 0$. Simply unrolling the recursion, we can obtain

$$a_n = \rho^n a_0 + b \left(\frac{1 - \rho^n}{1 - \rho}\right). \tag{29}$$

**Proof of the Theorem.**

Before we proceed, let us introduce some notation:

$$\begin{aligned}
G_i(\theta) &= \frac{\partial E}{\partial \theta}(\theta, X_i, Y_i), \\
A_{\varepsilon,i}(\theta) &= \{\varepsilon \le E(\theta, X_i, Y_i) - \tau(\theta) \le \varepsilon\}, \\
R_{\varepsilon,n}(\theta) &= \sum_{i=1}^{n} \chi_{A_{\varepsilon,i}(\theta)}, \\
S_{\epsilon,n}(\theta) &= \{i \in [n] : \chi_{A_{\epsilon,i}(\theta)} = 1\},
\end{aligned} \tag{30}$$

where $\chi_{A_{\varepsilon,i}(\theta)}$ is an indicator function for the event $A_{\varepsilon,i}(\theta)$, i.e.,

$$\chi_{A_{\varepsilon,i}(\theta)} = \begin{cases} 1, & \text{if } |E_\theta(X_i, Y_i) - \tau(\theta)| \le \varepsilon \\ 0, & \text{if } |E_\theta(X_i, Y_i) - \tau(\theta)| > \varepsilon \end{cases}. \tag{31}$$

We are now ready to analyze the estimator $\hat{\eta}_{\varepsilon,n}(\theta)$ for $\eta_\varepsilon(\theta)$:

$$\hat{\eta}_{\varepsilon,n}(\theta) = \begin{cases} \frac{1}{R_{\varepsilon,n}(\theta)} \sum_{i=1}^{n} \chi_{A_{\varepsilon,i}(\theta)} G_i(\theta), & \text{if } R_{\varepsilon,n}(\theta) > 0 \\ 0 & \text{if } R_{\varepsilon,n}(\theta) = 0 \end{cases}, \tag{32}$$

where $\varepsilon$ and $n$ are denoted explicitly to remove ambiguity.

Equipped with the basic results established earlier in this subsection, we can proceed first with proving assertion *(i)*. Note that, by definition (32), and because $\{X_i, Y_i\}_{i=1}^{n}$ are sampled independently, we have

$$\mathbb{E}\left[\frac{1}{|S|} \sum_{i \in S} G_i \,\Big|\, \bigcap_{i \in S} A_{\epsilon,i}(\theta)\right] = \frac{1}{|S|} \sum_{i \in S} \mathbb{E}\left[\chi_{A_{\epsilon,i}(\theta)} G_i(\theta) | A_{\epsilon,i}(\theta)\right] = \eta_\epsilon. \tag{33}$$

Also note that $S_{\epsilon,n}(\theta) = \emptyset$ is equivalent to $R_{\epsilon,n}(\theta) = 0$, and that

$$\mathbb{P}\left(S_{\epsilon,n}(\theta) = \emptyset\right) = \mathbb{P}\left(R_{\epsilon,n}(\theta) = 0\right) = q^n, \tag{34}$$

with $q = 1 - p$ and $p = \mathbb{P}\left(A_{\epsilon,i}(\theta)\right)$. Hence, we can get (i) as follows:

$$\mathbb{E}\left[\hat{\eta}_{\epsilon,n}(\theta)\right] = q^n \mathbb{E}\left[\hat{\eta}_{\epsilon,n}(\theta)|R_{\epsilon,n}(\theta) = 0\right] + (1 - q^n)\eta_\epsilon, \tag{35}$$

where we used the fact that $\sum_{S \neq \emptyset} \mathbb{P}\left(S_{\epsilon,n}(\theta) = S\right) = 1 - \mathbb{P}\left(S_{\epsilon,n}(\theta) = \emptyset\right) = 1 - q^n$.

Now, we prove (ii). We start by analyzing $\mathbb{E}\left[\hat{\eta}_{\epsilon,n}(\theta)\hat{\eta}_{\epsilon,n}(\theta)^\top\right]$:

$$
\begin{aligned}
\mathbb{E}\left[\hat{\eta}_{\epsilon,n}(\theta)\hat{\eta}_{\epsilon,n}(\theta)^\top\right] &= \sum_{S \subseteq [n]} \mathbb{P}\left(S_{\epsilon,n}(\theta) = S\right) \mathbb{E}\left[\hat{\eta}_{\epsilon,n}(\theta)\hat{\eta}_{\epsilon,n}(\theta)^\top|S_{\epsilon,n}(\theta) = S\right] \\
&= \mathbb{P}\left(S_{\epsilon,n}(\theta) = \emptyset\right) \mathbb{E}\left[\hat{\eta}_{\epsilon,n}(\theta)\hat{\eta}_{\epsilon,n}(\theta)^\top|S_{\epsilon,n}(\theta) = S_{\epsilon,n}(\theta)\right] \\
&\quad + \sum_{S \neq \emptyset} \mathbb{P}\left(S_{\epsilon,n}(\theta) = S\right) \mathbb{E}\left[\left(\frac{1}{|S|}\sum_{i \in S} G_i(\theta)\right)\left(\frac{1}{|S|}\sum_{i \in S} G_i(\theta)\right)^\top \Big| \bigcap_{i \in S} A_{\epsilon,i}(\theta)\right] \\
&= \mathbb{P}\left(S_{\epsilon,n}(\theta) = \emptyset\right) \mathbb{E}\left[\hat{\eta}_{\epsilon,n}(\theta)\hat{\eta}_{\epsilon,n}(\theta)^\top|R_{\epsilon,n}(\theta) = 0\right] \\
&\quad + \sum_{S \neq \emptyset} \mathbb{P}\left(S_{\epsilon,n}(\theta) = S\right) \frac{1}{|S|^2} \sum_{i \in S} \sum_{j \in S} \mathbb{E}\left[G_i(\theta)G_j(\theta)^\top|A_{\epsilon,i}(\theta), A_{\epsilon,j}(\theta)\right].
\end{aligned}
\tag{36}
$$

Now note that

$$
\begin{aligned}
\mathbb{E}\left[G_i(\theta)G_j(\theta)^\top|A_{\epsilon,i}(\theta), A_{\epsilon,j}(\theta)\right] &= \delta_{i,j}\mathbb{E}\left[G_i(\theta)G_i(\theta)^\top|A_{\epsilon,i}(\theta)\right] \\
&\quad + (1 - \delta_{ij})\mathbb{E}\left[G_i(\theta)|A_{\epsilon,i}(\theta)\right]\mathbb{E}\left[G_j(\theta)^\top|A_{\epsilon,j}(\theta)\right] \\
&= \mathbb{E}\left[G_i(\theta)|A_{\epsilon,i}(\theta)\right]\mathbb{E}\left[G_j(\theta)^\top|A_{\epsilon,i}(\theta)\right] \\
&\quad + \delta_{ij}(\mathbb{E}\left[G_i(\theta)G_i(\theta)^\top|A_{\epsilon,i}(\theta)\right] \\
&\quad - \mathbb{E}\left[G_i(\theta)|A_{\epsilon,i}(\theta)\right]\mathbb{E}\left[G_i(\theta)^\top|A_{\epsilon,i}(\theta)\right]) \\
&= \eta_\epsilon \eta_\epsilon^\top + \delta_{ij}\Sigma_\epsilon,
\end{aligned}
\tag{37}
$$

where we used the fact that $\{X_i, Y_i\}_{i=1}^n$ are sampled i.i.d. and the definitions in (30). Now, we can proceed as follows:

$$
\begin{aligned}
\mathbb{E}\left[\hat{\eta}_{\epsilon,n}(\theta)\hat{\eta}_{\epsilon,n}(\theta)^\top\right] &= q^n \mathbb{E}\left[\hat{\eta}_{\epsilon,n}(\theta)\hat{\eta}_{\epsilon,n}(\theta)^\top|R_{\epsilon,n}(\theta) = 0\right] \\
&\quad + \sum_{S \neq \emptyset} \frac{\mathbb{P}\left(S_{\epsilon,n}(\theta) = S\right)}{|S|}\left(|S|\eta_\epsilon \eta_\epsilon^\top + \Sigma_\epsilon\right) \\
&= q^n \mathbb{E}\left[\hat{\eta}_{\epsilon,n}(\theta)\hat{\eta}_{\epsilon,n}(\theta)^\top|R_{\epsilon,n}(\theta) = 0\right] \\
&\quad + (1 - q^n)\eta_\epsilon \eta_\epsilon^\top + f_n\Sigma_\epsilon,
\end{aligned}
\tag{38}
$$

where we write

$$f_n = \sum_{S \neq \emptyset} \frac{\mathbb{P}\left(S_{\epsilon,n}(\theta) = S\right)}{|S|}. \tag{39}$$

Now, we will show that

$$f_n \leq \frac{2 - p}{pn}. \tag{40}$$

First, let us define the following function

$$f(k) = \begin{cases} 0, & \text{if } k = 0 \\ \frac{1}{k} & \text{if } k \geq 1 \end{cases}, \tag{41}$$

and note that

$$f_n = \mathbb{E}\left[f(|S_{\epsilon,n}(\theta)|)\right] = \mathbb{E}\left[f(R_{\epsilon,n}(\theta))\right]. \tag{42}$$

Now note that

$$
\begin{aligned}
f_{n+1} &= \mathbb{E}\left[f(R_{\epsilon,n+1}(\theta))\right] \\
&= \mathbb{P}\left(A_{\epsilon,i}(\theta)^c\right)\mathbb{E}\left[f(R_{\epsilon,n}(\theta))\right] + \mathbb{P}\left(A_{\epsilon,i}(\theta)\right)\mathbb{E}\left[f(1 + R_{\epsilon,n}(\theta))\right] \\
&= q\mathbb{E}\left[f(R_{\epsilon,n}(\theta))\right] + p\mathbb{E}\left[f(1 + R_{\epsilon,n}(\theta))\right] \\
&= qf_n + p\mathbb{E}\left[\frac{1}{1 + R_{\epsilon,n}(\theta)}\right] \\
&= qf_n + \frac{1 - q^{n+1}}{n+1},
\end{aligned}
\tag{43}
$$

where, in the last equation, we used the fact shown in the preliminaries (see (28)):

$$
\mathbb{E}\left[\frac{1}{1 + R_{\epsilon,n}(\theta)}\right] = \frac{1 - q^{n+1}}{p(n+1)}.
\tag{44}
$$

Now let $a_n = nf_n$. We can write

$$
(n+1)f_{n+1} = (n+1)\left(qf_n + \frac{1 - q^n}{n+1}\right),
\tag{45}
$$

from which we obtain the following recursion:

$$
\begin{aligned}
a_{n+1} &= qnf_n + qf_n + (1 - q^{n+1}) \\
&= qa_n + qf_n + (1 - q^{n+1}) \\
&\leq qa_n + 1 + q,
\end{aligned}
\tag{46}
$$

where we used the fact that $f_n \leq 1$ and that $1 - q^n \leq 1$. With this recursion, we can now use the result illustrated in the preliminaries in (29) and get, using $q = 1 - p$,

$$
a_n \leq q^n a_0 + \frac{1 - q^n}{1 - q}(1 + q) \leq \frac{2 - p}{p}.
\tag{47}
$$

From the above inequality, we can conclude that

$$
0 \leq f_n = \frac{a_n}{n} \leq \frac{2 - p}{pn}.
\tag{48}
$$

Plugging this last result in (38), we can get

$$
\mathbb{E}\left[\hat{\eta}_{\epsilon,n}(\theta)\hat{\eta}_{\epsilon,n}(\theta)^\top\right] \preceq q^n\mathbb{E}\left[\hat{\eta}_{\epsilon,n}(\theta)\hat{\eta}_{\epsilon,n}(\theta)^\top | R_{\epsilon,n}(\theta) = 0\right] + (1 - q^n)\eta_\epsilon\eta_\epsilon^\top + \frac{2 - p}{pn}\Sigma_\epsilon.
\tag{49}
$$

We are now in the position to write and bound $\mathrm{cov}(\hat{\eta}_{\epsilon,n}(\theta))$:

$$
\begin{aligned}
\mathrm{cov}\left(\hat{\eta}_{\epsilon,n}(\theta)\right) &= \mathbb{E}\left[\hat{\eta}_{\epsilon,n}(\theta)\hat{\eta}_{\epsilon,n}(\theta)^\top\right] - \left[\hat{\eta}_{\epsilon,n}(\theta)\right]\left[\hat{\eta}_{\epsilon,n}(\theta)^\top\right] \\
&\preceq q^n\mathbb{E}\left[\hat{\eta}_{\epsilon,n}(\theta)\hat{\eta}_{\epsilon,n}(\theta)^\top | R_{\epsilon,n}(\theta) = 0\right] + (1 - q^n)\eta_\epsilon\eta_\epsilon^\top + \frac{2 - p}{pn}\Sigma_\epsilon \\
&\quad - \left(q^n\mathbb{E}\left[\hat{\eta}_{\epsilon,n}(\theta) | R_{\epsilon,n}(\theta) = 0\right] + (1 - q^n)\eta_\epsilon\right) \\
&\quad \cdot \left(q^n\mathbb{E}\left[\hat{\eta}_{\epsilon,n}(\theta) | R_{\epsilon,n}(\theta) = 0\right] + (1 - q^n)\eta_\epsilon\right)^\top \\
&= \frac{2 - p}{pn}\Sigma_\epsilon + (1 - q^n)\eta_\epsilon\eta_\epsilon^\top - (1 - q^n)^2\eta_\epsilon\eta_\epsilon^\top \\
&= \frac{2 - p}{pn}\Sigma_\epsilon + (1 - q^n)(1 - (1 - q^n))\eta_\epsilon\eta_\epsilon^\top \\
&= \frac{2 - p}{pn}\Sigma_\epsilon + (1 - q^n)q^n\eta_\epsilon\eta_\epsilon^\top \\
&\preceq \frac{2 - p}{pn}\Sigma_\epsilon + q^n\eta_\epsilon\eta_\epsilon^\top,
\end{aligned}
\tag{50}
$$

where we used (i), the fact that $1 - q^n \leq 1$ and the fact that $\mathbb{E}\left[\hat{\eta}_{\epsilon,n}(\theta) | R_{\epsilon,n}(\theta) = 0\right] = 0$, which follows by (32).

## B USEFUL FACTS AND DERIVATIONS

In this appendix, we provide, for completeness, some useful facts and explicit derivations of properties that we use in the paper. In particular, we show how the ordered statistics $E(\theta, X_{\omega_j(\theta)}, Y_{\omega_j(\theta)})$ in equation (5) are differentiable almost surely (with probability 1), and we explicitly derive equation (8) using the generalize chain rule (GCR).

### B.1 DIFFERENTIABILITY OF $E(\theta, X_{\omega_j(\theta)}, Y_{\omega_j(\theta)})$.

We will formally show that the ordered statistics $E_{(j)}(\theta)$, with $j = 1, ..., n$, are differentiable for any $\theta$ with probability 1. We first recall some notation. Let $E_{(1)}(\theta) \leq \ldots \leq E_{(n)}(\theta)$ denote the order statistics corresponding to the scalar random variables $E(\theta, X_1, Y_1), \ldots, E(\theta, X_n, Y_n)$.
Let us also denote by $\omega(\theta) : [n] \to [n]$ the permutation of indices $[n] := \{1, \ldots, n\}$ that correspond to the order statistics, i.e., $\omega(\theta) = (\omega_1(\theta), \ldots, \omega_n(\theta))$, and $(E_{(1)}(\theta), \ldots E_{(n)}(\theta)) = (E(\theta, X_{\omega_1(\theta)}, Y_{\omega_1(\theta)}), \ldots, E(\theta, X_{\omega_n(\theta)}, Y_{\omega_n(\theta)}))$. Now define the set $A_n$ as follows:

$$A_n = \{(E_1, ..., E_n) : E_i = E_j \text{ for some } i \neq j\}. \tag{51}$$

Now note that, by definition, the conformity score function $E(\theta, X, Y)$ is continuous and differentiable in $\theta$. Now fix some $\bar{\theta}$. Consider the event in which the ordered statistics are such that $E_{(1)}(\bar{\theta}) < \ldots < E_{(n)}(\bar{\theta})$, hence

$$(E(\bar{\theta}, X_{\omega_1(\bar{\theta})}, Y_{\omega_1(\bar{\theta})}), \ldots, E(\bar{\theta}, X_{\omega_n(\bar{\theta})}, Y_{\omega_n(\bar{\theta})})) = (E_{(1)}(\bar{\theta}), \ldots, E_{(n)}(\bar{\theta})) \notin A_n, \tag{52}$$

which means that $\omega(\bar{\theta})$ is the unique ordered statistics permutation for $\{E(\theta, X_i, Y_i)\}_{i=1}^n$ and note that this happens *almost surely* (with probability 1), because $E(\theta, X, Y)$ is a continuous function in $\theta$ and $X$. The key step is now to note that by continuity of $E(\theta, X_i, Y_i)$ in $\theta$, there exists $\delta > 0$ such that, for $\theta \in \{\theta' : \|\theta' - \bar{\theta}\| \leq \delta\}$, we have $\omega(\theta) = \omega(\bar{\theta})$, which means that, if $\|\theta - \bar{\theta}\| \leq \delta$,

$$\begin{aligned} (E_{(1)}(\theta), \ldots, E_{(n)}(\theta)) &= (E(\theta, X_{\omega_1(\theta)}, Y_{\omega_1(\theta)}), \ldots, E(\theta, X_{\omega_n(\theta)}, Y_{\omega_n(\theta)})) \\ &= (E(\theta, X_{\omega_1(\bar{\theta})}, Y_{\omega_1(\bar{\theta})}), \ldots, E(\theta, X_{\omega_n(\bar{\theta})}, Y_{\omega_n(\bar{\theta})})) \end{aligned} \tag{53}$$

At this point, let $j \in \{1, ..., n\}$, and let us denote $E_{(j)}(\theta) = E(\theta, X_{\omega_j(\theta)}, Y_{\omega_j(\theta)}) = E(\theta, \omega_j(\theta))$, and, for any $\theta \in \{\theta' : \|\theta' - \bar{\theta}\| \leq \delta\}$ the derivative of $E_{(j)}(\theta)$ is

$$\frac{\partial}{\partial \theta} E_{(j)}(\theta) = \frac{\partial}{\partial \theta} E(\theta, \omega_j(\theta)) = \frac{\partial}{\partial \theta} E(\theta, \omega_j(\bar{\theta})) = \frac{\partial E}{\partial \theta}(\theta, \omega_j(\bar{\theta})), \tag{54}$$

which is true because, as we show in (53) above, for $\theta \in \{\theta' : \|\theta' - \bar{\theta}\| \leq \delta\}$, the function $\omega_j(\theta)$ is a constant equal to $\omega_j(\bar{\theta})$. Note that, as we do in the main paper, we here denote by $\frac{\partial E}{\partial \theta}(\theta, \omega_j(\bar{\theta}))$ the partial derivative with respect to $\theta$. Note that, given that the choice of $\bar{\theta}$ is arbitrary, we have shown that the function $\theta \mapsto E(\theta, X_{\omega_j(\theta)}, Y_{\omega_j(\theta)})$ is indeed differentiable with probability 1 for all $j = 1, \ldots, n$.
To be absolutely convinced that (53) is true, note that we can show it by continuity of $\theta \mapsto E(\theta, X, Y)$, as follows: let's fix $\bar{\theta}$ and let us denote again $E_{(j)}(\theta) = E(\theta, X_{\omega_j(\theta)}, Y_{\omega_j(\theta)}) = E(\theta, \omega_j(\theta))$. We want to show that there exists $\delta > 0$ such that $\omega(\bar{\theta}) = \omega(\theta)$ for any $\theta \in \{\theta' : \|\theta' - \bar{\theta}\| \leq \delta\}$. To do so, it is sufficient to show that, for any $\theta \in \{\theta' : \|\theta' - \bar{\theta}\| \leq \delta\}$,

$$E(\theta, \omega_{i+1}(\bar{\theta})) > E(\theta, \omega_i(\bar{\theta})), \text{ for } i = 1, ..., n - 1. \tag{55}$$

Let us define

$$\varepsilon = \min_{i=1,\ldots,n-1} \{E(\bar{\theta}, \omega_{i+1}(\bar{\theta})) - E(\bar{\theta}, \omega_i(\bar{\theta}))\}. \tag{56}$$

From continuity of $\theta \mapsto E(\theta, X, Y)$, there exists $\delta > 0$ such that if $\theta \in \{\theta' : \|\theta' - \bar{\theta}\| \leq \delta\}$, we have

$$|E(\theta, \omega_i(\bar{\theta})) - E(\bar{\theta}, \omega_i(\bar{\theta}))| < \frac{\epsilon}{2}. \tag{57}$$

Note that, from (56) and (57), we have for all $i = 1, \ldots, n$,

$$E(\bar{\theta}, \omega_{i+1}(\bar{\theta})) \geq E(\bar{\theta}, \omega_i(\bar{\theta})) + \epsilon, \tag{58}$$

and

$$E(\theta, \omega_i(\bar{\theta})) > E(\bar{\theta}, \omega_i(\bar{\theta})) - \frac{\epsilon}{2}. \tag{59}$$

Hence, note that, starting from this last inequality, and then using (58)

$$\begin{aligned} E(\theta, \omega_{i+1}(\bar{\theta})) &> E(\bar{\theta}, \omega_{i+1}(\bar{\theta})) - \frac{\epsilon}{2} \\ &\geq E(\bar{\theta}, \omega_i(\bar{\theta})) + \frac{\epsilon}{2}. \end{aligned} \tag{60}$$

Now, we can use again (57) (continuity) to show that

$$E(\bar{\theta}, \omega_i(\bar{\theta})) > E(\theta, \omega_i(\bar{\theta})) - \frac{\epsilon}{2}, \tag{61}$$

and thus observe that, for all $i = 1, \ldots, n$,

$$E(\theta, \omega_{i+1}(\bar{\theta})) > E(\bar{\theta}, \omega_i(\bar{\theta})) + \frac{\epsilon}{2} > E(\theta, \omega_i(\bar{\theta})), \tag{62}$$

from which we can confirm that, for $\theta \in \{\theta' : \|\theta' - \bar{\theta}\| \leq \delta\}$, $\omega(\theta) = \omega(\bar{\theta})$, and we can conclude.

## B.2 Explicit derivation of equation (8)

Please note that equation (8) follows from taking the derivative of a function of multiple variables and the chain rule. This is also called the *generalized chain rule* in some textbooks (Herman & Strang, 2018)(see Theorem 4.10). In the paper, when writing

$$\frac{\partial}{\partial \theta} \ell(\theta, \hat{\tau}(\theta), X, Y), \tag{63}$$

we mean the *total* derivative of the function $\theta \mapsto l(\theta, \hat{\tau}(\theta), X, Y)$, evaluated at a dummy $\theta$. On the other hand, when writing

$$\frac{\partial \ell}{\partial \theta}(\theta, \hat{\tau}(\theta), x, y), \tag{64}$$

we mean the *partial* derivative of $\ell(\theta, q, x, y)$ with respect to $\theta$, evaluated at $(\theta, q, x, y) = (\theta, \hat{\tau}(\theta), X, Y)$. The difference is that, in the partial derivative, $\hat{\tau}(\theta)$ is treated as a constant, whereas for the total derivative we do not treat $\hat{\tau}(\theta)$ as a constant. Now, the generalized chain rule (in vector form) can be written as follows: let $u(\theta) \in \mathbb{R}^n$ and $v(\theta) \in \mathbb{R}^m$ be two differentiable functions of $\theta$, and $f(u, v)$ a differentiable function of two vector variables $u$ and $v$. Then

$$\frac{\partial}{\partial \theta} f(u(\theta), v(\theta)) = \left(\frac{\partial u}{\partial \theta}(\theta)\right)^\top \frac{\partial f}{\partial u}(u(\theta), v(\theta)) + \left(\frac{\partial v}{\partial \theta}(\theta)\right)^\top \frac{\partial f}{\partial v}(u(\theta), v(\theta)), \tag{65}$$

where $\frac{\partial u}{\partial \theta}(\theta)$ is the Jacobian of $u(\theta)$, i.e., the matrix with $\frac{\partial u_i}{\partial \theta_j}(\theta)$ in the $i$-th row and $j$-th column (equivalently, $\frac{\partial v}{\partial \theta}(\theta)$ is the Jacobian of $v(\theta)$). Note that in the case of $\ell(\theta, \hat{\tau}(\theta), x, y)$, $x$ and $y$ do not depend on $\theta$ so we can focus on $\ell$ as a function of the two functions $u(\theta) = \theta$ and $v(\theta) = \hat{\tau}(\theta)$. Replacing these $u(\theta)$ and $v(\theta)$ in equation (65), and replacing $f(u(\theta), v(\theta))$ with $\ell(\theta, \hat{\tau}(\theta), x, y)$ we see that then

$$\frac{\partial}{\partial \theta} \ell(\theta, \hat{\tau}(\theta), x, y) = \frac{\partial \ell}{\partial \theta}(\theta, \hat{\tau}(\theta), x, y) + \frac{\partial \ell}{\partial \hat{\tau}}(\theta, \hat{\tau}(\theta), x, y) \frac{\partial \hat{\tau}}{\partial \theta}(\theta), \tag{66}$$

which is precisely equation (8) in the main paper, where we used the fact that $\left(\frac{\partial \theta}{\partial \theta}\right) = I_d$, where $I_d$ is a $d \times d$ identity matrix, with $d$ the dimension of $\theta$.

Given that usually in textbooks the generalized chain rule (GCR) is only shown for scalar multi-variable functions, we now report the derivation of equation (8) using the scalar GCR as reported and proved in the statement of Theorem 4.10 in (Herman & Strang, 2018). Hence, we will now provide the derivation of (8) at a more granular level. Consider a differentiable function $\ell$ of $k$ variables, $\ell : \mathbb{R}^k \to \mathbb{R}$. Now let $f_1, ..., f_k$ be differentiable functions, with $f_i : \mathbb{R}^d \to \mathbb{R}$, for $i = 1, ..., k$ and some $d \geq 1$. Then, denoting a vector $[t_1, ..., t_d] \in \mathbb{R}^d$ and $w = \ell(f_1(t_1, ..., t_d), ..., f_k(t_1, ..., t_d))$ we have (GCR):

$$\frac{\partial w}{\partial t_j} = \sum_{i=1}^{k} \frac{\partial w}{\partial f_i} \frac{\partial f_i}{\partial t_j}. \tag{67}$$

Now note that in the case of our paper, we have $w = \ell(\theta, \hat{\tau}(\theta), x, y)$. Note that $x$ and $y$ have no dependency on parameters in $\theta$ and hence their derivatives will be zero. We can then focus on $\theta$ and $\hat{\tau}(\theta)$. For convenience, note that we can write $\theta = [\theta_1, ..., \theta_d]$. Now note that the gradient of $w$ is

$$\frac{\partial}{\partial \theta}[w] = \left[\frac{\partial w}{\partial \theta_1}, ..., \frac{\partial w}{\partial \theta_d}\right]^\top. \tag{68}$$

Now note that, for some $j \in \{1, ..., d\}$, using the chain rule (67) above,

$$\begin{aligned} \frac{\partial w}{\partial \theta_j} &= \sum_{i=1}^{d} \frac{\partial w}{\partial \theta_i}\frac{\partial \theta_i}{\partial \theta_j} + \frac{\partial \ell}{\partial \hat{\tau}}(\theta, \hat{\tau}(\theta), x, y)\frac{\partial \hat{\tau}}{\partial \theta_j}(\theta) \\ &\quad + \frac{\partial w}{\partial x}\frac{\partial x}{\partial \theta_j} + \frac{\partial w}{\partial y}\frac{\partial y}{\partial \theta_j} \\ &= \frac{\partial \ell}{\partial \theta_j}(\theta, \hat{\tau}(\theta), x, y) + \frac{\partial \ell}{\partial \hat{\tau}}(\theta, \hat{\tau}(\theta), x, y)\frac{\partial \hat{\tau}}{\partial \theta_j}(\theta), \end{aligned} \tag{69}$$

where we used the fact that $\frac{\partial \theta_i}{\partial \theta_j} = 0$ if $i \neq j$ and $\frac{\partial \theta_i}{\partial \theta_i} = 1$. We also explicitly used the fact that $\frac{\partial x}{\partial \theta_j} = 0$ and $\frac{\partial y}{\partial \theta_j} = 0$ because the samples do not depend on the parameter $\theta$. Stacking together $\frac{\partial w}{\partial \theta_j}$ we can see that we obtain precisely equation (8) of the paper:

$$\begin{aligned} \frac{\partial}{\partial \theta}[w] &= \frac{\partial}{\partial \theta}[\ell(\theta, \hat{\tau}(\theta), X, Y)] \\ &= \frac{\partial \ell}{\partial \theta}(\theta, \hat{\tau}(\theta), X, Y) + \frac{\partial \ell}{\partial \hat{\tau}}(\theta, \hat{\tau}(\theta), X, Y)\frac{\partial \hat{\tau}}{\partial \theta}(\theta). \end{aligned} \tag{70}$$

## C  ADDITIONAL EXPERIMENTS

Here, we provide additional experimental results to complement the findings in the main paper.

### C.1  ADDITIONAL TRAINING CURVES

We first present additional training curves, specifically the test loss and accuracy per epoch, for each dataset. These plots highlights the performance throughout the training process, providing further insights into convergence behavior and generalization performance. It can be seen that the test loss exhibits a pattern similar to the training loss in 3. In terms of accuracy, VR-ConfTr achieves higher accuracy than ConfTr.

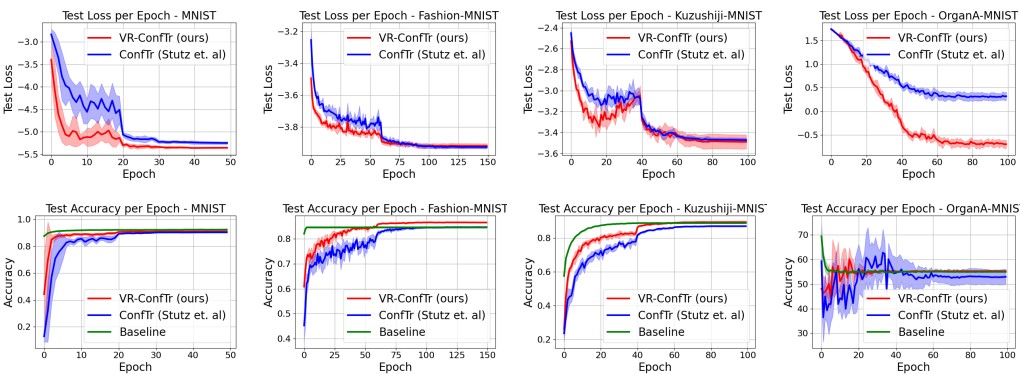

Figure 4: Training curves for MNIST, Fashion-MNIST, Kuzushiji-MNIST, and OrganAMNIST. For each dataset, we show the test loss on the first row and tets accuracy on the bottom row at the end of each epoch.

#### C.1.1  VARIANCE OF THE GRADIENTS OVER THE COURSE OF TRAINING

In this section, we present visualization of the variance of the estimated quantile gradients during training for our proposed method Vr-ConfTr, compared to ConfTr in figure 5. We conduct this experiment on the MNIST dataset, using the $m$-ranking estimator with Vr-ConfTr, and evaluate performance across different batch sizes. This analysis aims to empirically substantiate our claim that Vr-ConfTr reduces variance of the estimated quantile gradients over the epochs, leading to more stable gradient updates and improved final performance. Furthermore, we demonstrate that with an appropriate choice of the hyperparameter $m$ for the $m$-ranking estimator, Vr-ConfTr not only reduces variance but also shows improvements in terms of the bias of the estimated quantile gradients during training. In order to compute the variance and bias for the estimated quantile gradient $\widehat{\frac{\partial \tau}{\partial \theta}}$, we estimate the population quantile $\tau(\theta)$ and its gradient $\frac{\partial \tau}{\partial \theta}$ at each model update utilizing the full training, calibration, and test datasets.

### C.2  ABLATION STUDY FOR $m$ AND $\varepsilon$

#### C.2.1  $\varepsilon$-THRESHOLD ESTIMATOR ABLATION STUDY

This study evaluates the bias and variance of the $\widehat{\frac{\partial \tau}{\partial \theta}}$ using the $\varepsilon$-threshold estimator with Vr-ConfTr for the GMM dataset depicted in figure 2. Figure 6 shows how varying $\varepsilon$ impacts the estimator's performance, highlighting the trade-offs between bias and variance of $\widehat{\frac{\partial \tau}{\partial \theta}}$ as $\varepsilon$ changes.

#### C.2.2  $m$-RANKING ESTIMATOR ABLATION STUDY

We evaluate the bias and variance of $\widehat{\frac{\partial \tau}{\partial \theta}}$ using the $m$-ranking estimator with Vr-ConfTr for the GMM dataset. Figure 7 shows how varying $m$ impacts the estimator's performance, highlighting the

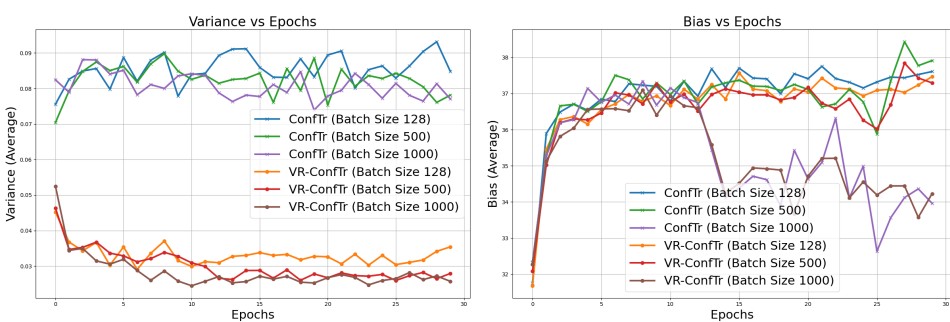

Figure 5: Variance and bias of the estimated quantile gradients during training for ConfTr and Vr-ConfTr, evaluated on the MNIST dataset across different batch sizes. The left figure shows the variance of the gradients over epochs. The right panel illustrates the bias of the estimated gradients, demonstrating that Vr-ConfTr maintains low bias while effectively reducing variance.

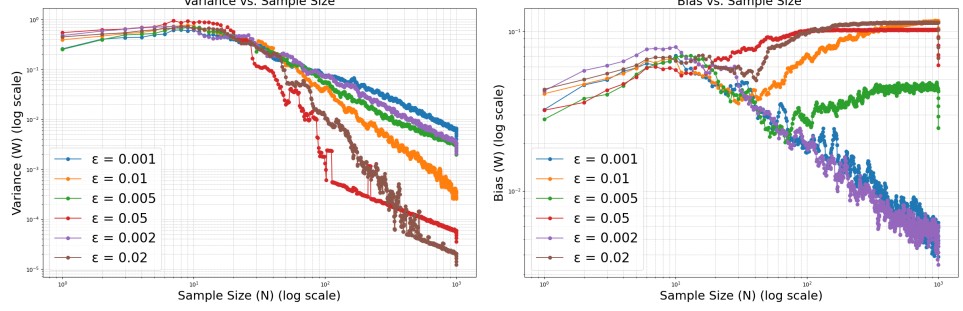

Figure 6: Bias and variance for the quantile gradient estimates using the $\varepsilon$-threshold estimator with Vr-ConfTr on the GMM dataset. The left panel shows the variance, and the right panel shows the bias for different $\varepsilon$ values.

trade-offs between bias and variance of $\widehat{\frac{\partial \tau}{\partial \theta}}$ as $m$ changes. Here $m$ explicitly depends on the desired miscoverage rate $\alpha$ and the sample size $n$.

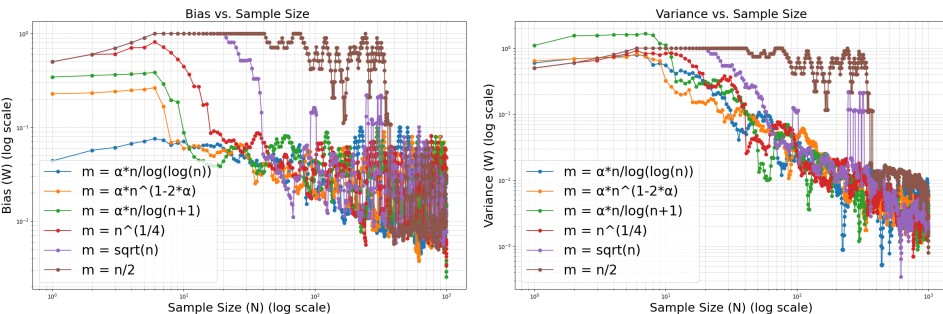

Figure 7: Bias and variance for the quantile gradient estimate using the $m$-ranking estimator with `Vr-ConfTr` on the GMM dataset. The left panel shows the bias, and the right panel shows the variance for different $m$ values

### C.2.3 ON THE CONNECTION BETWEEN $\epsilon$-THRESHOLD AND $m$-RANKING ESTIMATORS

As mentioned at the end of section 3.1, the $m$-ranking and the $\varepsilon$-threshold estimator are intimately related, and are indeed almost the same estimator. We may also say that the $m$-ranking estimator is a special case of the $\varepsilon$-threshold estimator in which $\varepsilon$ is chosen "adaptively" with respect to batch and parameter $\theta$ *via the integer* $m$. To see this, note that, for a calibration batch $\{X_i, Y_i\}_{i=1}^n$ with $n$ samples, fixing an integer $m$, the $m$-ranking estimator can be seen as the $\varepsilon$-threshold estimator with $\varepsilon = \inf\left\{\varepsilon' > 0 : \sum_{i=1}^n 1_{\hat{A}_{\varepsilon',i}(\theta)} \geq m\right\}$, where $\hat{A}_{\varepsilon,i}(\theta) = \{|E_\theta(X_i, Y_i) - \hat{\tau}(\theta)| \leq \varepsilon\}$ and $1_A$ is the indicator function for the event $A$: in words, $\varepsilon$ is the smallest value such that $m$ samples' conformity scores from the current calibration batch fall within $\varepsilon$-distance from $\hat{\tau}(\theta)$. We now explain why the $m$-ranking strategy is a natural choice as opposed to fixing $\varepsilon$ across all iterations. In practice, when training the models, we noticed that a "good" value of $\varepsilon$ *varies significantly* across iterations. Note that a good value of the threshold $\varepsilon$ not only depends on the specific batch $B_{\text{cal}}$ at a given iteration, but also on the model parameters $\theta$ at that iteration. Hence, hyper-parameter tuning with the $\varepsilon$-threshold estimator requires some heuristic to adapt the threshold to specific iterations. In this sense, the $m$-ranking estimator is a natural heuristic for a batch and parameter-dependent choice of the threshold $\varepsilon$. We noticed indeed that performing hyper-parameter tuning of the $m$-ranking estimator we were able to provide a good value of $m$ to be used *across all iterations*, which from the point of view of hyper-parameter tuning is a great advantage.

To empirically illustrate this connection and validate the importance of dynamically tuning the $\varepsilon$-threshold estimator, figure 8 presents the optimal adaptive tuning of the $\varepsilon$-threshold estimator on the Fashion-MNIST dataset. This tuning ensures that the $\varepsilon$-threshold estimator achieves comparable performance to the $m$-ranking estimator with $m = 6$, which was used to train the model.

### C.3 CLASS-CONDITIONAL COVERAGE AND SET SIZE

We evaluated the trained models in terms of class-conditional coverage and set size, using the same CP-procedure applied post-training with the standard `THR` method and $\alpha = 0.01$. Figure 9 displays the class-conditional coverage and set sizes for each dataset. The results show the effectiveness of `Vr-ConfTr` in achieving reliable class-conditional coverage with smaller class-conditional prediction set sizes. The results are taken as the average over all the training and testing trials to ensure robustness and reliability.

### C.4 TUNING VR-CONFTR: NUMBER OF POINTS FOR GRADIENT ESTIMATION (M)

In `VR-ConfTr`, the number of points ($m$) used to compute the gradient estimate plays a crucial role in the bias-variance trade-off. Consistent with the theory, increasing $m$ (which with the $\varepsilon$-threshold estimator would translate to increasing the threshold $\varepsilon$) reduces the variance but potentially increases

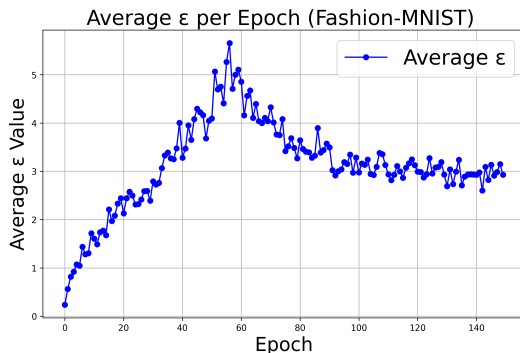

Figure 8: Adaptive tuning of the $\varepsilon$-threshold estimator on Fashion-MNIST. The plot shows the evolution of the threshold $\varepsilon$ across training iterations, required to match the $m$-ranking estimator with $m = 6$. The variability in $\varepsilon$ underscores the necessity of dynamic adjustment in threshold-based approaches.

the bias of the gradient estimate. We conduct a grid search over the values $[4, 6, 8, 10, 16, 20]$ for $m$ and report the results of tuning $m$ for MNIST, and Fashion MNIST, selecting the value of $m$ that experimentally provides the best trade-off between bias and variance. **MNIST Results.** As shown in Fig. 10, we observe a consistent reduction in the variance of gradient estimate as $m$ increases. However once we pass the optimal threshold the bias increases as can be seen by the higher values of the training loss as well as decrease in the size of the prediction sets. The figures corresponding to the loss on the training data per epoch, the loss on the test data per epoch, the accuracy evaluate on the test data per epoch, as well as the prediction set size evalauted on the test data per epoch.

**Fashion-MNIST.** Similarly tuning $m$ on Fashion-MNIST shows that a value of $m = 6$ provides the best results, as depicted in Fig. 11

### C.5 ALTERNATIVE ARCHITECTURE

In this section, we compare the performance of `VR-ConfTr` on Kushuniji-MNIST using a simpler linear model architecture. The results indicate that regardless of the model architecture, the trends observed in terms of convergence speed and prediction set efficiency are consistent across datasets and architectures. Table 2 shows the average accuracy and set sizes for the two different models trained on K-MNIST.

| Dataset | Model Name | Accuracy (Avg ± Std) | Set Size (Avg ± Std) |
|---|---|---|---|
| K-MNIST (Linear) | Baseline | $0.695 \pm 0.007$ | $6.799 \pm 0.117$ |
| | ConfTr | $0.582 \pm 0.047$ | $6.646 \pm 0.226$ |
| | VR-ConfTr | $0.612 \pm 0.033$ | $6.488 \pm 0.148$ |
| K-MNIST (MLP) | Baseline | $0.872 \pm 0.046$ | $4.982 \pm 0.530$ |
| | ConfTr | $0.783 \pm 0.125$ | $4.762 \pm 0.226$ |
| | VR-ConfTr | $0.835 \pm 0.098$ | $4.657 \pm 0.680$ |

Table 2: Evaluation results of the KMNIST dataset trained with different model architectures. Columns present average accuracy and set size with their standard deviations (**Avg ± Std**).

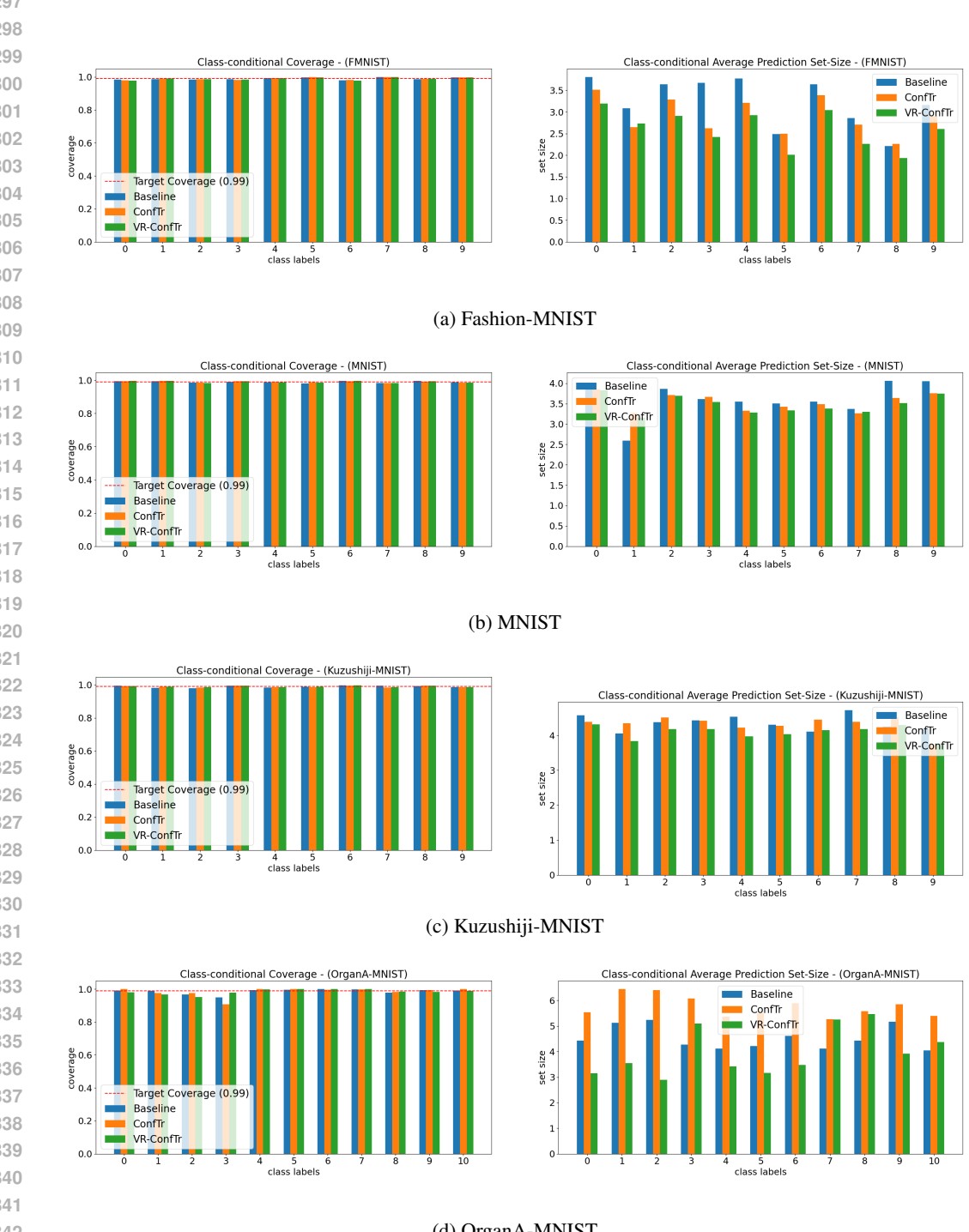

(a) Fashion-MNIST

(b) MNIST

(c) Kuzushiji-MNIST

(d) OrganA-MNIST

Figure 9: Class-conditional coverage rates and average prediction set sizes for each dataset, averaged over 10 test trials. For each dataset, the left plot shows the class-conditional coverage rates with the target coverage level of $1 - \alpha = 0.99$ indicated by the horizontal red dashed line. The right plot shows the class-conditional average prediction set sizes.

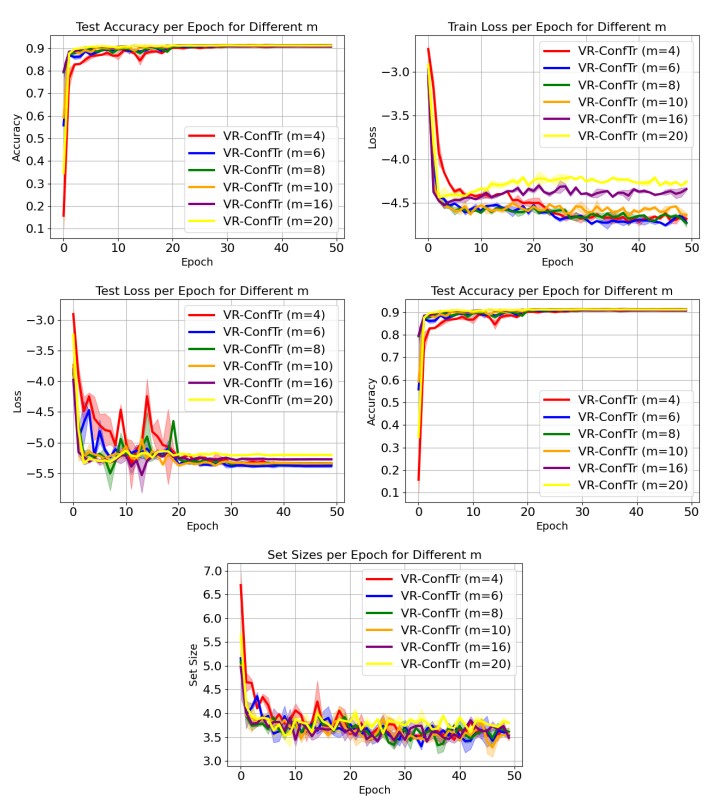

Figure 10: Training trajectories for different values of $m$ on MNIST data

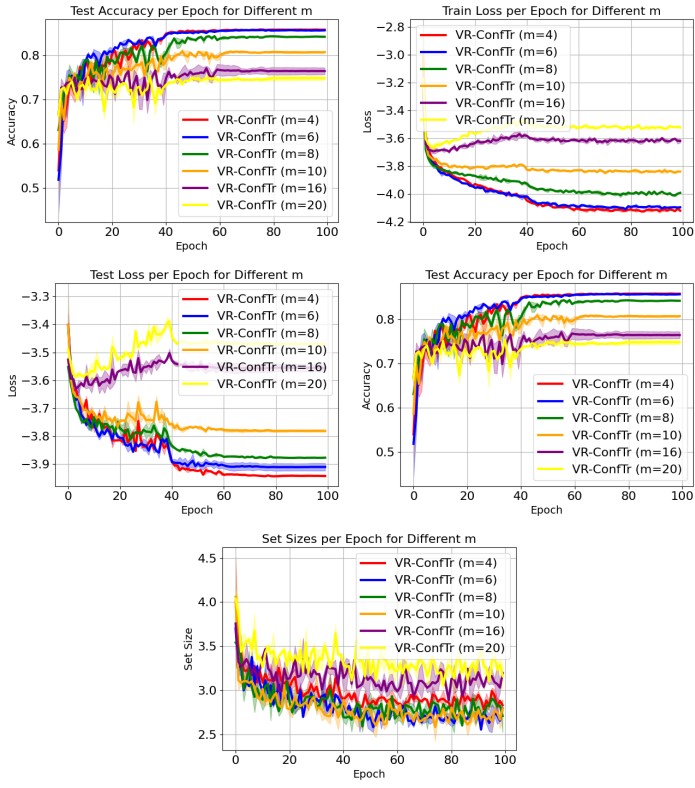

Figure 11: Training trajectories for different values of $m$ on Fashion-MNIST data

## D EXPERIMENTAL DETAILS

In this section we describe the experimental setup, including model architectures, dataset configurations, training protocol, testing procedure, and the corresponding hyper-parameters. The focus of the experiments is on evaluating the conformal prediction (CP) set sizes and ensuring a fair comparison between the baseline Conformal Training `ConfTr` and our proposed `VR-ConfTr`.

### D.1 DATASET CONFIGURATIONS

We consider the benchmark datasets MNIST LeCun et al. (1998), Fashion-MNIST Xiao et al. (2017b), Kuzushiji-MNIST Clanuwat et al. (2018) and OrganAMNIST Yang et al. (2021).MNIST is a dataset of handwritten digits with 10 classes, and Fashion-MNIST consists of 10 fashion product categories. Kuzushiji-MNIST extends the MNIST paradigm by incorporating 10 classes of cursive Japanese characters. OrganAMNIST, derived from medical images, contains 11 classes of abdominal organ slices. The training, calibration, and testing splits for each dataset are summarized in Table 3. MNIST and Fashion-MNIST are provided by the torchvision library, while Kuzushiji-MNIST and OrganAMNIST are available from their respective repositories. For MNIST, Fashion-MNIST, and Kuzushiji-MNIST, 10% of the training set is reserved as calibration data. For OrganAMNIST, the validation set is used as the calibration data. During evaluation, we combine the calibration and test data and perform evaluations over 10 random splits of the combined dataset into calibration/test partitions. Model parameters are learned exclusively on the training data, while calibration and test data are used to evaluate the model as a black-box at the end of each epoch. The transformations applied to the dataset are as follows: for MNIST, Fashion-MNIST, and Kuzushiji-MNIST, images are normalized to have zero mean and unit variance, using a mean of 0.5 and a standard deviation of 0.5. For OrganAMNIST, images undergo random horizontal flips, random rotations of up to 15 degrees, and are normalized similarly.

| Dataset | Classes | Image Size | Training Set | Calibration Set | Test Set |
|---|---|---|---|---|---|
| MNIST | 10 | $28 \times 28$ | 55,000 | 5,000 | 10,000 |
| Fashion-MNIST | 10 | $28 \times 28$ | 55,000 | 5,000 | 10,000 |
| OrganMNIST | 11 | $28 \times 28$ | 34,561 | 6,491 | 17,778 |
| Kuzushiji-MNIST | 10 | $28 \times 28$ | 55,000 | 5,000 | 10,000 |

Table 3: Dataset Splits

### D.2 MODEL ARCHITECTURES

In our experiments, we implemented all models using JAX Bradbury et al. (2018). We utilize a range of architectures including linear models, multi-layer perceptrons (MLPs), and modified ResNet architectures tailored for specific datasets. For the **MNIST** dataset, we employ a simple linear model, which consists of a single dense layer. The input images, reshaped from $28 \times 28$ into a flattened vector of size 784, are passed through a fully connected layer mapping the inputs directly to the 10 output classes. This architecture provided a minimalistic baseline for comparison. For **Fashion-MNIST**, we use a multi-layer perceptron (MLP), with two hidden layers. We use 64 units per hidden layer, with ReLU activations Nair & Hinton (2010) , followed by a dense layer for the 10 output classes. For **Kuzushiji-MNIST**, we utilize a similar MLP architecture. The model contains two hidden layers with 256 and 128 units, respectively. The input data is flattened and passed through these fully connected layers with ReLU activations. For **OrganAMNIST**, we used a residual network, inspired by the ResNet architecture from He et al. (2016) , with modifications. The model consists of an initial convolutional layer followed by four stages of residual blocks, each with two layers. Each residual block uses $3 \times 3$ convolutions with ReLU activations. The number of output channels doubles after each state $(64, 128, 256, 512)$. Global average pooling is applied before the final fully connected layer, which maps the pooled feature representations to the 11 output classes. We do not attempt to optimize the model architectures in order to solve the datasets with high accuracy. Instead, we focus on the conformal prediction results, and ensure that the architecture used across different algorithms are identical for a fair comparison.

## D.3 TRAINING DETAILS

Similar to Stutz et al. (2022), we trained all models using Stochastic Gradient Descent (SGD) with Nesterov momentum Sutskever et al. (2013). The learning rate follows a multi-step schedule where the initial learning rate was decreased by a factor of 0.1 after 2/5, 3/5, and 4/5 of the total number of epochs. The models were trained using cross-entropy-loss for `Baseline` training, and for `ConfTr` and `VR-ConfTr` based on the size-loss as described by Stutz et al. (2022). During training, we set the conformal prediction threshold parameter $\alpha = 0.01$. To ensure statistical robustness, we conducted multiple randomized training trials for each dataset, using a different random seed for each trial. Specifically, we performed 10 training trials for MNIST and 5 training trials each for FMNIST, KMNIST, and OrganAMNIST. During each trial, a unique random seed was used to initialize the model and optimizer, ensuring that each trial followed a distinct learning trajectory. The corresponding training trajectories, i.e the training loss, testing loss, accuracy and CP set sizes evaluated on the test data at the end of every epoch, were averaged over these randomized trials to provide a smooth and general view of the model's performance. The key hyper-parameters used for training are listed in Table 4. These hyper-parameters include **size weight** which scales the loss term associated with the size of the CP sets during training, **alpha** $\alpha$ corresponding to the miscoverage rate is set to 0.01. **batch size** for SGD, **learning rate** for the optimizer, and the number of **epochs** for which the model is trained for.

| Hyper-parameter | MNIST | Fashion-MNIST | Kuzushiji-MNIST | OrganA-MNIST |
|---|---|---|---|---|
| Batch Size | 500 | 500 | 500 | 500 |
| Training Epochs | 50 | 150 | 100 | 100 |
| Learning Rate | 0.05 | 0.01 | 0.01 | 0.01 |
| Optimizer | SGD | SGD | SGD | SGD |
| Temperature | 0.5 | 0.1 | 0.1 | 0.5 |
| Target Set Size | 1 | 0 | 1 | 1 |
| Regularizer Weight | 0.0005 | 0.0005 | 0.0005 | 0.0005 |
| Size Weight | 0.01 | 0.01 | 0.01 | 0.1 |
| Alpha ($\alpha$) | 0.01 | 0.01 | 0.01 | 0.01 |
| Num. of Pts for Gradient | 6 | 6 | 4 | 4 |

Table 4: Training and evaluation Hyper-parameters for each dataset.

## D.4 EVALUATION DETAILS

The evaluation of our models was conducted in two stages: (1) computing the test accuracy for each model after training, and (2) evaluating the conformal prediction (CP) set sizes and coverage over multiple test splits. The goal was to ensure both accuracy and conformal prediction performance are consistently reported across randomized trials and test splits. **Test Accuracy:** For each dataset, the test accuracy of the trained models was evaluated on the test data, and the results were averaged over the randomized training trials. **CP set sizes** To compute the average conformal prediction (CP) set size, we first combine the holdout calibration and test data. We then randomly split this combined data into calibration and test portions, repeating the process 10 times. For each split, we apply the CP `THR` algorithm with $\alpha = 0.01$ to compute the prediction set sizes on the test portion, and the results are averaged across the 10 random splits. The cardinality of each split is consistent with the dataset configurations outlined in Table 3. This procedure is performed for each trained model, and the final reported results are averaged across both the training trials and testing splits.

## D.5 DIFFERENCES FROM CONFTR REPORTS

We report the performance of `Conftr` with a batch size of 100 for Fashion-MNIST, as originally reported by Stutz et al. (2022), selected for optimal performance. While a batch size of 500 yields smaller set sizes, it results in a slight ( 1%) decrease in accuracy. For completeness, we include the results for both configurations. **Retrieving exact reported set sizes as Stutz et al. (2022)**: Our experimental results and trends align with those reported in Stutz et al. (2022). However, the smaller set sizes for `Conftr` on MNIST and FMNIST in their paper are likely due to their use

| Model | Batch Size | Accuracy (Avg ± Std) | Set Size (Avg ± Std) |
|---|---|---|---|
| ConfTr | 100 | $0.809 \pm 0.051$ | $3.125 \pm 0.197$ |
| ConfTr | 500 | $0.799 \pm 0.065$ | $3.048 \pm 0.201$ |
| VR-ConfTr | 500 | $0.839 \pm 0.043$ | $2.795 \pm 0.154$ |

Table 5: Final evaluation results for Fashion-MNIST, showing average accuracy and set size with their standard deviations (**Avg ± Std**).

of more advanced/different architectures. Despite this, the overall trends— `Conftr` outperforming `Baseline`, and `VR-Conftr` outperforming `Conftr`—remain consistent regardless of the model. Our focus is on a fair comparison across algorithms by using the same architecture, rather than reproducing the exact figures or architectures from Stutz et al. (2022).

# E    ON THE COMPUTATIONAL COMPLEXITY OF VR-CONFTR.

We will now discuss the computational complexity of VR-ConfTr when compared to ConfTr. We will argue that the computational complexity of the two algorithms is essentially the same. We start by breaking down the computational cost of ConfTr and then illustrate the difference with VR-ConfTr.

**Per-step computational complexity of ConfTr.** Given a batch and partition $B = \{B_{\text{cal}}, B_{\text{pred}}\}$, with $|B_{\text{cal}}| = |B_{\text{pred}}| = n$, the first step of ConfTr is to compute a sample $\alpha$ quantile $\hat{\tau}(\theta)$ based on the calibration batch $B_{\text{cal}} = \{X_i^{\text{cal}}, Y_i^{\text{cal}}\}_{i=1}^n$, which requires the computation of the calibration batch conformity scores $\{E_\theta(X_i^{\text{cal}}, Y_i^{\text{cal}})\}_{i=1}^n$ and of their $\alpha$-quantile. At this point, the computation of the ConfTr gradient is performed computing the gradient of the loss

$$\frac{1}{|B_{\text{pred}}|} \sum_{(x,y) \in B_{\text{pred}}} \ell(\theta, \hat{\tau}(\theta), x, y). \tag{71}$$

Note that for each sample $(x, y)$, computing the ConfTr gradient implies computing the following (equation (8) in the main paper):

$$\frac{\partial}{\partial \theta}[\ell(\theta, \hat{\tau}(\theta), x, y)] = \frac{\partial \ell}{\partial \theta}(\theta, \hat{\tau}(\theta), x, y) + \frac{\partial \ell}{\partial \hat{\tau}}(\theta, \hat{\tau}(\theta), x, y)\frac{\partial \hat{\tau}}{\partial \theta}(\theta). \tag{72}$$

Note that computing this gradient requires computing (i) the gradients $\frac{\partial \ell}{\partial \theta}(\theta, \hat{\tau}(\theta), x, y)$ and $\frac{\partial \ell}{\partial \tau}(\theta, \hat{\tau}(\theta), x, y)$ for all samples $(x, y) \in B_{\text{cal}}$, and (ii) the gradient $\frac{\partial \hat{\tau}}{\partial \theta}(\theta)$. The difference in terms of computational complexity between ConfTr and our proposed VR-ConfTr lies in the computation of estimates of $\frac{\partial \tau}{\partial \theta}(\theta)$, which in ConfTr is done via computing the gradient of $\hat{\tau}(\theta)$, while in our algorithm is done plugging an improved estimate $\widehat{\frac{\partial \tau}{\partial \theta}}(\theta)$. We describe the computational difference between these two approaches in the next paragraph.

**Per-step computational complexity of VR-ConfTr.**   Note that in our proposed algorithm VR-ConfTr, given a batch $B$ defined as above, we consider the same per-step loss function of ConfTr of equation (71). However, instead of computing directly the gradient of (71), we compute separately an estimate $\widehat{\frac{\partial \tau}{\partial \theta}}(\theta)$ of $\frac{\partial \tau}{\partial \theta}(\theta)$ using our novel estimation technique and then plug this estimate in equation (72) in place of $\frac{\partial \hat{\tau}}{\partial \theta}(\theta)$. In the proposed estimator, computing $\widehat{\frac{\partial \tau}{\partial \theta}}(\theta)$ equals computing gradients $\{\frac{\partial E}{\partial \theta}(\theta, x, y)\}_{(x,y) \in \bar{B}}$, where $\bar{B}$ is the set containing the $m$ samples whose conformity scores fall within $\epsilon$ distance from the sample quantile $\hat{\tau}(\theta)$, or the $m$ samples whose conformity scores are the closest to $\hat{\tau}(\theta)$ in the case of the $m$-ranking estimator. Note that, computationally, our algorithm requires computing $\frac{\partial \ell}{\partial \theta}(\theta, \hat{\tau}(\theta), x, y)$ and $\frac{\partial \ell}{\partial \tau}(\theta, \hat{\tau}(\theta), x, y)$, which is the same as ConfTr, while we do not need to compute the gradient $\frac{\partial \hat{\tau}}{\partial \theta}(\theta)$. Instead, we replace the computation of the gradient of $\hat{\tau}(\theta)$ with the computation of an average of $m$ gradients of conformity scores. Note that, while the computational complexity of our estimate $\widehat{\frac{\partial \tau}{\partial \theta}}(\theta)$ is clear and it is $m$ times the complexity of computing $\frac{\partial E}{\partial \theta}(\theta, x, y)$, the computational complexity of computing $\frac{\partial \hat{\tau}}{\partial \theta}(\theta)$ depends on the specific technique adopted to compute the gradient of a sample quantile. The most basic version is the one we discuss in equation (5) in the main paper, which would involve the computation and average of the gradients of two conformity scores. However, note that in practice the authors of ConfTr declare that they use smooth sorting to compute the sample quantile $\hat{\tau}(\theta)$ - and this is consistent with what we observe in their publicly released code. Crucially, differentiating a sample quantile obtained via smooth sorting potentially involves the computation of the gradients of all the samples in the batch $B_{\text{cal}}$, because smooth sorting - as implemented byStutz et al. (2022) - creates functional dependencies between the conformity scores of all samples in the calibration batch. In conclusion, the main computational difference between ConfTr and VR-ConfTr is in the computation of the estimate of $\frac{\partial \tau}{\partial \theta}(\theta)$, which for both of the techniques boils down to computing and averaging a certain set of conformity scores. This is why we can safely conclude that the computational complexity of the two algorithms is essentially the same.

