# OpenReview forum: "Conformal Training with Reduced Variance"
_ICLR.cc/2025/Conference — Submitted to ICLR 2025_

### Official Review · Reviewer_YP6a · 2024-10-27

**Soundness:** 3
**Presentation:** 3
**Contribution:** 3
**Rating:** 5
**Confidence:** 3

**Summary:**

This paper proposes a new approach, Variance-Reduced Conformal Training, which adds a variance reduction in the estimation of gradients to improve the efficiency of conformal prediction during training. It is an attempt to sort out the inefficiency problems that appear with other methods like ConfTr, for which the estimation of gradients is problematic because of noise in the data batches. These improvements in the proposal have brought both an increase in the speed of convergence and the efficiency in the size of the sets of predictions, as confirmed by experiments using different benchmark datasets. This effectively preserves probabilistic guarantees to obtain much more compact and reliable prediction sets when compared to baseline models.

**Strengths:**

1. VR-ConfTr consistently generates smaller prediction sets with better length efficiency than its predecessor, ConfTr, and baseline models.

2. The paper provides a solid theoretical analysis to support the variance reduction claims, offering insights into the bias-variance trade-off involved.

3. The paper is well-written and clearly structured.

**Weaknesses:**

1. VR-ConfTr relies on a large calibration set to accurately estimate the quantile estimator $\hat{\tau}$ and its gradient $\hat{\frac{\partial \tau}{\partial \theta}}$, which are critical for variance reduction.

2. While VR-ConfTr improves prediction set efficiency, it introduces additional computational steps, which may slow down training in scenarios with limited computational resources.

**Questions:**

1. How does VR-ConfTr handle situations with limited/imbalanced calibration data, and how does this affect coverage guarantees?

2. The paper mainly focuses on MNIST-type datasets, which restricts the demonstration of generality and robustness. How would VR-ConfTr perform on high-resolution images?

---

> ### Author Response · Authors · 2024-11-22
>
> Thank you for your review. We will address all your concerns below:
>
> **Reviewer:**
>
> 1. VR-ConfTr relies on a large calibration set to accurately estimate the quantile estimator
> $\hat{\tau}$ and its gradient $\hat{\frac{\partial \tau}{\partial \theta}}$, which are critical for variance reduction.
>
> **Authors:**
>
> Please note that a sufficiently large calibration batch size is one of the requirements for the original ConfTr algorithm. Indeed, the primary motivation behind VR-ConfTr and one of the key findings of our work is that ConfTr not only relies on large calibration batches, but also *fails to efficiently utilize the available data samples during training*. In this direction, VR-ConfTr takes a major step towards effectively optimizing models during training according to conformal prediction efficiency metrics. This is achieved thanks to our novel "plug-in" approach with which we can replace the sample quantile gradient ${\frac{\partial \hat\tau}{\partial \theta}}$ with an improved, novel and provably sample efficient estimator $\widehat{\frac{\partial \tau}{\partial \theta}}$ of the population quantile $\frac{\partial \tau}{\partial \theta}(\theta)$
>
> **Reviewer:**
>
> 2. While VR-ConfTr improves prediction set efficiency, it introduces additional computational steps, which may slow down training in scenarios with limited computational resources.
>
> **Authors:**
>
> Note that the computational cost of our proposed VR-ConfTr is essentially the same as ConfTr. In fact, our proposed "plug-in" technique consists in replacing a specific component in the gradient estimation pipeline of the ConfTr algorithm, which is the estimation of the population quantile gradient, $\frac{\partial \tau}{\partial \theta}(\theta)$. Indeed, by differentiating the objective directly as a function of the sample quantile $\hat{\tau}(\theta)$, ConfTr ends up using the gradient of the sample quantile $\frac{\partial \hat{\tau}}{\partial \theta}(\theta)$. As one of the main contributions of this paper, we show that this specific estimator is sample inefficient, and our proposed VR-ConfTr is specifically designed to replace it - thanks to our "plug-in" technique - with a better, provably sample-efficient estimator $\widehat{\frac{\partial {\tau}}{\partial \theta}}$. In particular, we adopt the $m$-ranking estimator, which can be seen as a special case of the $\epsilon$-threshold estimator, which, as we show analytically, provably reduces the estimation variance with the calibration batch size, i.e., it is sample-efficient. Remarkably, the computational complexity of this estimator is essentially the same as computing $\frac{\partial \hat{\tau}}{\partial \theta}(\theta)$ - which is what is done in ConfTr - because it just boils down to computing and averaging $m$ derivatives of conformity scores. The other parts in the estimation of the objective function gradient have exactly the same computational complexity for both algorithms, and thus we can conclude that the computational complexity of the two algorithms is essentially the same.
> We provide a more detailed description of this reasoning in a new dedicated Appendix E in the revised version of the paper.

---

> ### Author Response · Authors · 2024-11-22
>
> **Reviewer:**
>
> Q1: How does VR-ConfTr handle situations with limited/imbalanced calibration data, and how does this affect coverage guarantees?
>
>
> **Authors:**
>
> Please note that, as we remarked above, the main contribution of our work is our approach to effectively overcome the sample-inefficiency of the existing ConfTr  algorithm. Hence, effectively using the samples available in the calibration batch when simulating conformal prediction in-between training updates is precisely the sharp advancement that we provide and the
> key motivation for our research effort. In order to be effective, VR-ConfTr does need a sufficient number of samples to be used to simulate conformal prediction in-between training updates. However, we remark that our result is precisely to show that we can effectively improve the sample-efficiency over the ConfTr benchmark, as we show analytically and empirically. We concur though that the point raised by the reviewer is relevant and further studies on these cases - with limited availability of training samples to be used to simulate conformal prediction during training - are an interesting venue for future work.
>
> **Reviewer:**
>
> Q2: The paper mainly focuses on MNIST-type datasets, which restricts the demonstration of generality and robustness. How would VR-ConfTr perform on high-resolution images?
>
> **Authors:**
>
> With respect to datasets with high resolution images, note that our key contribution in this paper is to detect and provide an effective solution to the sample inefficiency problem of the original ConfTr algorithm. This is a critical problem ***even with benchmark datasets*** (with low resolution images) ***and relatively small models***. We believe that our work is therefore a significant first step towards the possibility of applying conformal training techniques to large scale models and datasets. The experiments with the medical dataset OrganA-MNIST and the clear superior performance and effectiveness of VR-ConfTr as compared to ConfTr with this relatively more complex dataset are promising in this direction. We believe that hyper-parameter tuning needed for higher resolution images datasets and large-scale models necessitates extensive investigation and it is beyond the scope of this paper.

---

> > ### Author Response · Authors · 2024-11-26
> > **Follow-up request**
> >
> > Dear reviewer, thank you again for your review. Please let us know if you have any follow-up questions or comments, we would be happy to address any further concern or question you might have on our paper and responses.

---

> > > ### Author Response · Authors · 2024-11-27
> > > **Follow-up**
> > >
> > > Dear reviewer, thank you again for your review. Given that this is the last day to update the revised manuscript, we would like to take the opportunity to ask you for follow-up comments or questions that might help us improve the paper.
> > > We are also happy to elaborate more in the upcoming discussion days for any other queries or comments on paper and responses.

---

> > > > ### Author Response · Authors · 2024-12-02
> > > > **Final follow-up**
> > > >
> > > > Dear reviewer, we greatly appreciate your feedback. Given that this is the last day for us to receive your responses, please let us know if we were able to address all your concerns.

---

### Official Review · Reviewer_eSTW · 2024-11-04

**Soundness:** 1
**Presentation:** 2
**Contribution:** 3
**Rating:** 3
**Confidence:** 4

**Summary:**

The authors propose a new training objective for optimizing for conformal predictive set size, based on the work of (Stutz et al., 2022). By focusing on the computation of the gradients of the loss rather than the loss itself, the authors identify an alternative estimator of the conformal threshold to be incorporated in the loss that is more sample efficient, leading to less noisy gradients, and thus more stable training, faster convergence and better models.

**Strengths:**

The problem of building models that optimize for conformal predictive set size is a natural one, and yet often left to the sides. Indeed, there are still few works that tackle this problem, and more work on the topic is certainly welcome. The proposed solution is novel and can be of use to a wide audience.

**Weaknesses:**

I have two major issues with the paper:
1. The mathematical reasoning in the paper does not seem sound
2. Even if they were, the theoretical analysis seems lacking.
Given these two points, the paper could still be held up by a comprehensive empirical analysis. However, while there is some empirical analysis, it is hardly comprehensive, and could be improved on (especially if it is needed in order to compensate for (1.) and (2.)).

On the soundness of the mathematical reasoning:
- The derivatives in equation (5) do not exist. As currently written, the chain rule cannot be applied. Perhaps it can be salvaged with subgradients, though.
- I was unable to understand why (8) holds, or where it comes from.

Lacking theoretical analysis: the authors only prove their reduced variance with one particular estimator, which does not even match the estimator used in the experiments; moreover, this was done under significant simplifying assumptions. If these assumptions were so light as the authors imply, then it should be easy to prove a relaxed version of the theorem that incorporates them. (Change-of-measure inequalities may be useful here.)
This is to the point that I barely consider the paper to have any theoretical analysis.

On the experiments:
- The current visualizations are almost exclusively learning curves. While their presence is welcome, there are probably better ways to visualize, e.g., the variance of the gradients over the course of the training, which the authors claim to have reduced. From the learning curves, it is not entirely clear that these have really been reduced.
- It would be nice to also investigate the conditional coverage of the models. Does using your training objective lead to better conditional coverage? (Even if the answer is 'no', it would still be nice to have this in the supplementary material, and would not count negatively, in my view.)
- The authors should consider more datasets and models -- currently only CNNs on MNIST-like datasets. I suggest the authors also consider:
  - ViTs, still on image data
  - Transformers (e.g., BERT-style) on text classification tasks
  - GNNs for graph-based tasks, e.g., node prediction
  Since the paper proposes a new training objective, in the absence of a better theoretical understanding of it (possibly including on its interactions with gradient descent methods), it is important to test it on a wider variety of architectures.

There are also some minor issues:
- The use of $\min\_{\theta \in \Theta} L(\theta) := [\cdots]$ can be a bit confusing, as it seems at a first glance that the equality is claimed with regards to the minimum. A better way to write this is $\min\_{\theta \in \Theta} L(\theta) \quad \text{where}\ L(\theta) := [\cdots]$.
- In line 317, I suggest avoiding the use of the term 'gradient-boosting estimators'; gradient boosting estimators are an established class of ML models (e.g., XGBoost, LightGBM, etc.). These have little to do with the estimators being constructed in the paper, and caused a fair amount of head-scratching on my end. How about just calling these 'estimators for $\eta(\theta)$'?
- The writing in the proof of lemma 3.1 is rather unorthodox in the context of a paper. Consider avoiding long chains of {in,}equalities, especially, when you feel the need to follow it up with explanations for why the steps of the chain hold (e.g., line 682).
- On tables 1, 2 and 5, use the standard $\mu \pm \sigma$ notation in a single column for average and standard deviation, rather than separating into multiple columns.
- The notation is overall very heavy, probably unnecessarily so. This makes the paper a bit hard to parse.

Finally, it is worth noting that over half of the paper is spent effectively describing the work of (Stutz et al., 2022). Perhaps this can be trimmed somewhat, or better streamlined.

**Questions:**

My main questions:

- Some clarification on (8) would be appreciated. Why does it hold? The text claims that it comes from differentiating $\ell$, but that does not make sense. Is there a typo?
- How does the derivation of the loss fare in the face of the non-existant derivatives in (5)? (Or do they actually exist and I've missed something?)
- How does the proposed method fare when using the estimator in equation (13) rather than the ranking estimator?

**Score:** the weaknesses (particularly those relating to soundness and the theoretical analysis) seem rather significant to me, making me lean towards rejection. Given their severity, I do not think the rejection is borderline, even if the experimental results seem somewhat positive. Should the reviewers improve their presentation and soundness of their theoretical results, I would be willing to increase my score.

---

> ### Author Response · Authors · 2024-11-22
>
> We thank the reviewer for their insightful feedback. We will address all your concerns below:
>
> **Reviewer:**
>
> 1. The mathematical reasoning in the paper does not seem sound
> 2. Even if they were, the theoretical analysis seems lacking. Given these two points, the paper could still be held up by a comprehensive empirical analysis. However, while there is some empirical analysis, it is hardly comprehensive, and could be improved on (especially if it is needed in order to compensate for (1.) and (2.)).
>
> **Authors:**
>
> Thank you for your feedback. We provide our detailed responses regarding both soundness and meaningfulness of our theoretical analysis below where we address your concerns.
>
> **Reviewer:**
> On the soundness of the mathematical reasoning:
>
> The derivatives in equation (5) do not exist. As currently written, the chain rule cannot be applied. Perhaps it can be salvaged with subgradients, though.
>
> **Authors:**
>
> Thank you for focusing your attention on equation (5), we are very happy to address this concern, which also allows us to enrich and improve the clarity in the manuscript. We note that the derivatives in equation (5) do, in fact, exist with probability 1. This is true because (i) the conformity score function $\theta\mapsto E(\theta, X, Y)$ is continuous and differentiable in $\theta$, and continuous in $X$; (ii) samples $(X_i, Y_i)$ are sampled independently, and hence the probability of the event that two conformity scores are identical is zero; (iii) as a consequence of the previous point, the order statistics are $E_{(1)}(\theta) < \ldots < E_{(n)}(\theta)$ - notice the strict inequalities - with probability 1, and hence, for any fixed $\bar{\theta}$, continuity of $\theta\mapsto E(\theta, X, Y)$ guarantees that the permutations of indices $\omega(\theta)$ that correspond to the order statistics is such that $\omega(\theta) = \omega(\bar\theta)$ for $\theta$ in a sufficiently small neighborhood of $\bar\theta$, and thus $E(\theta, X_{\omega_{j}(\theta)}, Y_{\omega_{j}(\theta)}) = E(\theta, X_{\omega_{j}(\bar\theta)}, Y_{\omega_{j}(\bar\theta)})$. Hence, for any fixed $\bar\theta$ the map $\theta\mapsto E(\theta, X_{\omega_{j}(\theta)}, Y_{\omega_{j}(\theta)})$ is a composition of differentiable functions, and it is therefore differentiable, in the neighborhood of $\bar\theta$. Given that the choice of $\bar\theta$ is arbitrary, $\theta\mapsto E(\theta, X_{\omega_{j}(\theta)}, Y_{\omega_{j}(\theta)})$ is differentiable everywhere with probability 1.
>
> We have added a note about the almost sure (probability 1) differentiability of $E(\theta, X_{\omega_{j}(\theta)}, Y_{\omega_{j}(\theta)})$ after equation (5) in the revised paper. For completeness, we have included a detailed and rigorous derivation of the differentiability of $E(\theta, X_{\omega_{j}(\theta)}, Y_{\omega_{j}(\theta)})$ in the revised paper, in the new Appendix B, namely "Useful Facts and Derivations" - specifically Appendix B.1. We hope that this high level explanation together with the detailed rigorous derivation in the appendix are satisfactory, and that the reviewer is fully convinced. Otherwise, we are happy to discuss further.

---

> ### Author Response · Authors · 2024-11-22
>
> **Reviewer:**
> I was unable to understand why (8) holds, or where it comes from.
>
> **Authors:**
>
> We apologize if the notation of this equation was not clear. Please note that the equation follows from taking the derivative of a function of multiple variables and the chain rule. This is also called the *generalized chain rule* in some textbooks (for example, see Theorem 4.10 in [R1]: Herman and Strand, *Calculus, vol 3*, 2018). In the paper, when writing
> \begin{equation}
> \frac{\partial}{\partial \theta}\ell(\theta, \hat{\tau}(\theta), X, Y),
> \end{equation}
> we mean the derivative/gradient, over $\theta$, of the function $\ell$, which is a function of multiple variables, which are $(\theta, \hat{\tau}(\theta), X, Y)$. When writing
> \begin{equation}
> \frac{\partial \ell}{\partial \theta}(\theta, \hat{\tau}(\theta), x, y),
> \end{equation}
> instead, we mean the partial derivative of $\ell$ with respect to the first variable/argument $\theta$.
> Now, the generalized chain rule (in vector form) can be written as follows: let $u(\theta)\in\mathbb{R}^n$ and $v(\theta)\in\mathbb{R}^m$ be two differentiable functions of $\theta$, and $f(u,v)$ a differentiable function of two vector variables $u$ and $v$. Then
>
> \begin{equation}
> \frac{\partial }{\partial \theta}f(u(\theta), v(\theta)) = \left(\frac{\partial u}{\partial \theta}(\theta)\right)^\top \frac{\partial f}{\partial u}(u(\theta), v(\theta)) + \left(\frac{\partial v}{\partial \theta}(\theta)\right)^\top \frac{\partial f}{\partial v}(u(\theta), v(\theta)),
> \end{equation}
>
> where $\frac{\partial u}{\partial \theta}(\theta)$ is the Jacobian of $u(\theta)$, i.e., the matrix with $\frac{\partial u_i}{\partial \theta_j}(\theta)$ in the $i$-th row and $j$-th column (equivalently, $\frac{\partial v}{\partial \theta}(\theta)$ is the Jacobian of $v(\theta)$). Note that in the case of $\ell(\theta, \hat{\tau}(\theta), x, y)$, $x$ and $y$ do not depend on $\theta$ so we can focus on $\ell$ as a function of the two functions $u(\theta) = \theta$ and $v(\theta) = \hat{\tau}(\theta)$. Replacing these $u(\theta)$ and $v(\theta)$ in equation above, and replacing $f(u(\theta), v(\theta))$ with $\ell(\theta, \hat{\tau}(\theta), x, y)$ we see that then
> \begin{equation}
> \frac{\partial}{\partial \theta}\ell(\theta, \hat{\tau}(\theta), x, y) = \frac{\partial \ell}{\partial \theta}(\theta, \hat{\tau}(\theta), x, y) + \frac{\partial \ell}{\partial \hat{\tau}}(\theta, \hat{\tau}(\theta), x, y)\frac{\partial \hat{\tau}}{\partial \theta}(\theta),
> \end{equation}
> which is precisely equation (8) in the submitted manuscript, where we used the fact that $\left(\frac{\partial \theta}{\partial \theta}\right) = I_d$, where $I_d$ is a $d\times d$ identity matrix, with $d$ the dimension of $\theta$.
>
> We have included the above description and an additional derivation at a more granular level in the new Appendix B, "Useful Facts and Derivations" in the revised manuscript - specifically Appendix B.2, for completeness. We hope that the reviewer will find this satisfactory, and that the soundness of equation (8) and (9) is fully clarified. We are happy to discuss further otherwise.
>
> [R1]: Herman and Strand, \emph{Calculus, vol 3}, openstax, 2018 (freely available at https://openstax.org/details/books/calculus-volume-3/)

---

> > ### Author Response · Authors · 2024-11-22
> >
> > **Reviewer:**
> >
> > Lacking theoretical analysis: the authors only prove their reduced variance with one particular estimator, which does not even match the estimator used in the experiments; moreover, this was done under significant simplifying assumptions. If these assumptions were so light as the authors imply, then it should be easy to prove a relaxed version of the theorem that incorporates them. (Change-of-measure inequalities may be useful here.) This is to the point that I barely consider the paper to have any theoretical analysis.
> >
> > **Authors:**
> >
> > On the $\varepsilon$-threshold and $m$-ranking estimators:
> >
> > Please note that, as we argue at the end of section 3.1, the $m$-ranking and the $\varepsilon$-threshold estimator are intimately related, and are indeed almost the same estimator. We may also say that the $m$-ranking estimator is a special case of the $\varepsilon$-threshold estimator in which $\varepsilon$ is chosen "adaptively" with respect to the batch and parameter $\theta$ *via the integer $m$*. To see this, note that, for a calibration batch $[X_i, Y_i]_{i=1}^n$ with $n$ samples, fixing an integer $m$, the $m$-ranking estimator can be seen as the $\varepsilon$-threshold estimator with $\varepsilon$ the smallest threshold such that $m$ samples' conformity scores from the current calibration batch fall within $\varepsilon$-distance from $\hat{\tau}(\theta)$. We now explain why the $m$-ranking strategy is a natural choice as opposed to fixing $\varepsilon$ across all iterations. In practice, when training the models, we noticed that a "good" value of $\varepsilon$ *varies significantly* across iterations. Note that a good value of the threshold $\varepsilon$ not only depends on the specific calibration batch at a given iteration, but also on the model parameters $\theta$ at that iteration. Hence, hyper-parameter tuning with the $\varepsilon$-threshold estimator requires some heuristic to adapt the threshold to specific iterations. In this sense, the $m$-ranking estimator is a natural heuristic for a batch and parameter-dependent choice of the threshold $\varepsilon$. We noticed indeed that performing hyper-parameter tuning of the $m$-ranking estimator we were able to provide a good value of $m$ to be used *across all iterations*, which from the point of view of hyper-parameter tuning is a great advantage. Thanks to your comment, we realize we could have better underlined this aspect and provided more discussions and insights on the connection between the $\varepsilon$-threshold and the $m$-ranking estimators, and on why the $m$-ranking is a good choice for hyper-parameter tuning. We include more discussions on this in the revised paper. In addition, please note that both estimators
> > are developed over the theoretical result of Lemma 1, which is the main fundamental pillar for both our novel analysis of the ConfTr variance, and for the study of the novel estimators.
> >
> > As an additional remark, we would also like to emphasize that our main contribution is detecting for the first time a main source of sampling inefficiency in the ConfTr algorithm, and developing an effective solution based on plugging better estimates of the population quantile gradient in the ConfTr objective function gradient. In this last step, we propose and analyze a candidate provably sample-efficient estimator and find an effective implementation of this estimator in the $m$-ranking approach. We agree that further theoretical analyses are possible and we think our work is a first step in a very new and growing area of model learning/optimization (which we can refer to as conformal training, but which we also mathematically formulate as conformal risk minimization). Hence, although our work leaves some theoretical questions open, we believe it is and will be very valuable for the community.
> >
> > **On the assumption $\hat{\tau}(\theta) = \tau(\theta)$.**
> >
> > Please note that the sample quantile $\hat{\tau}$ is an asymptotically unbiased estimator of the population quantile and the bias is asymptotically normal (see for example Van der Vaart, A. W. (1998), *Asymptotic Statistics*, Cambridge University Press, Chapter 21). Hence, for a sufficiently large calibration batch size, it's very reasonable to assume that the value of $\hat{\tau}(\theta)$ will be very close to $\tau(\theta)$. At the same time, assuming a sufficiently large calibration batch size to compute a good estimate of the population quantile is a fundamental requirement of ConfTr, and our focus is instead on the fact that the large calibration batch size *is not enough to provide a sample efficient gradient estimation*, observation from which we develop our novel solution based on our plug-in technique and variance reduction approach. At the same time, we concur with the reviewer that further theoretical investigations would be valuable and we plan to conduct them as future work.

---

> > > ### Author Response · Authors · 2024-11-22
> > >
> > > **Reviewer:** On the experiments:
> > >
> > > 1. The current visualizations are almost exclusively learning curves. While their presence is welcome, there are probably better ways to visualize, e.g., the variance of the gradients over the course of the training, which the authors claim to have reduced. From the learning curves, it is not entirely clear that these have really been reduced.
> > >
> > > **Authors:**
> > >
> > > We thank the reviewer for their insightful feedback regarding the visualization of gradient variance during training. In response to your suggestion, we have added a dedicated section in appendix C.1.1 of the revised manuscript.
> > > In this new section, we provide visualizations of the variance of the estimated quantile gradients using the $m$-ranking estimator with our proposed method Vr-ConfTr, as well as ConfTr across epochs. In addition to epochs, we also show different batch sizes to provide a comprehensive analysis. The results further validate our claim that Vr-ConfTr effectively reduces the variance of the estimated quantile gradients during training.
> > >
> > > **Reviewer:**
> > >
> > > 2. It would be nice to also investigate the conditional coverage of the models. Does using your training objective lead to better conditional coverage? (Even if the answer is 'no', it would still be nice to have this in the supplementary material, and would not count negatively, in my view.)
> > >
> > > **Authors:**
> > >
> > > Thank you for raising this interesting point. To investigate this, we performed an additional empirical study, evaluating the class conditional coverage and set sizes on the final trained models. Our findings indicate that VR-ConfTr does, indeed, lead to better conditional coverage - in most cases. The experiment shows that, similarly to what happens for marginal coverage, VR-ConfTr improves over ConfTr in terms of length-efficiency when evaluated on each class. The results for all 4 datasets and the experimental details have been added to the appendix C.3 in the revised paper. We report both the class conditional coverage and the class conditional set sizes.
> > >
> > >
> > > **Reviewer:**
> > >
> > > 3. The authors should consider more datasets and models -- currently only CNNs on MNIST-like datasets. I suggest the authors also consider:
> > > - ViTs, still on image data
> > > - Transformers (e.g., BERT-style) on text classification tasks
> > > - GNNs for graph-based tasks, e.g., node prediction Since the paper proposes a new training objective, in the absence of a better theoretical understanding of it (possibly including on its interactions with gradient descent methods), it is important to test it on a wider variety of architectures.
> > >
> > > **Authors:**
> > >
> > > Thank you for these valuable suggestions. We concur with the reviewer that experimenting with large datasets and models will further validate the effectiveness of conformal training and conformal risk minimization approaches, including the sample-efficiency advantages of our VR-ConfTr solution. We note, however, that our key contribution in this paper is to detect and provide an effective solution to the sample inefficiency problem of the original ConfTr algorithm, which was a problem *even with benchmark datasets and relatively small models*. We believe that our work is therefore a significant first step towards the possibility of applying conformal training techniques to large scale models and datasets. The experiments with the medical dataset OrganA-MNIST and the clear superior performance and effectiveness of VR-ConfTr as compared to ConfTr  with this relatively more complex dataset is promising in this direction.

---

> > > > ### Author Response · Authors · 2024-11-22
> > > > **Regarding Minor issues**
> > > >
> > > > We thank you for your feedback on these minor issues. We have revised the manuscript accordingly to address all of them.  We hope you find the revisions satisfactory. Please let us know if you have any further minor issues.

---

> > > > > ### Author Response · Authors · 2024-11-22
> > > > >
> > > > > **Reviewer:**
> > > > >
> > > > > Finally, it is worth noting that over half of the paper is spent effectively describing the work of (Stutz et al., 2022). Perhaps this can be trimmed somewhat, or better streamlined.
> > > > >
> > > > > **Authors:**
> > > > >
> > > > > Thank you for your feedback. We respectfully note that the effective description of Conftr is
> > > > > only subsection 2.2, which is around 1 page. The remaining of Section 2 is first the (novel) formulation of conformal risk minimization (CRM) and then the analysis of the variance of the gradient of Conftr, which is novel and instrumental to introduce our solution to the sample-inefficiency problem. To make the role of Section 2 clearer, we have decided to change the title of Section 2 from "Conformal Training" to "Problem Formulation", which we believe clarifies the role of the section and considerably improves the structure and flow of the paper. Please let us know if you have any further comments on this.
> > > > >
> > > > >
> > > > > ***Questions***
> > > > >
> > > > > **Reviewer:**
> > > > >
> > > > > Some clarification on (8) would be appreciated. Why does it hold? The text claims that it comes from differentiating $\ell$, but that does not make sense. Is there a typo?
> > > > >
> > > > > **Authors:**
> > > > >
> > > > > Please see our response to your comment above, and the more detailed rigorous illustration in the new Appendix B of the paper: "Useful Facts and Derivations". Please let us know if you would like us to provide further details/clarifications.
> > > > >
> > > > > **Reviewer:**
> > > > >
> > > > > How does the derivation of the loss fare in the face of the non-existant derivatives in (5)? (Or do they actually exist and I've missed something?)
> > > > >
> > > > > **Authors:**
> > > > >
> > > > > Please see our response to your comment above. The derivatives do, in fact, exist with probability 1. We illustrate the reasoning at a high level in the response to the comment above and we flesh out all the details in a rigorous derivation in the new Appendix B, "Useful Facts and Derivations". Please let us know if you would like further details/clarifications.
> > > > >
> > > > > **Reviewer:**
> > > > >
> > > > > How does the proposed method fare when using the estimator in equation (13) rather than the ranking estimator?
> > > > >
> > > > > **Authors:**
> > > > >
> > > > > Please see our response to the above comment related to the intimate connection between the $m$-ranking and the $\varepsilon$-threshold estimators, and the fact that the $m$-ranking estimator is essentially a special case of the $\varepsilon$-threshold estimator. What we noted experimentally is that a good choice of $\varepsilon$ changes significantly across iterations, making it hard to find a single good hyper-parameter $\varepsilon$. A potential choice to effectively adapt $\varepsilon$ during training is the $m$-ranking estimator. The $m$-ranking estimator can indeed be seen as a special case of the $\varepsilon$-estimator in which $\varepsilon$ is chosen as the smallest value for which $m$ samples' conformity scores fall within $\varepsilon$ distance from $\hat{\tau}(\theta)$.
> > > > > To empirically illustrate this connection, we have included an additional study validating the importance of dynamically tuning $\varepsilon$, demonstrating how the threshold $\varepsilon$ containing $m$ conformity scores significantly changes across epochs. We hope section C.2 in the appendix fully addresses the reviewer's question.

---

> ### Comment · Reviewer_eSTW · 2024-11-25
>
> **Derivatives that do not exist:** the justification given in the rebuttal is very unsatisfactory (and the new appendix B.1 in the paper is not much better). First, continuity and iid do not imply that ties happen with probability 0. Second, unless I'm mistaken, almost sure differentiability is not enough to guarantee the validity of the chain rule, which is the key property used in the derivation in the paper. This is not a minor issue. \
> Now, I'll be honest, and *please* forgive me if I go too overboard or am incorrect here: reading this part of the rebuttal felt almost like reading mathematical nonsense. The reasoning was very reminiscent of the usual outputs of LLMs when doing mathematical proofs, in that at the same time it seemed very flawed, but with just a tint of plausibility that forced me not to dismiss it outright. And, to be honest, many bits of the original submission already gave me a slight LLM ick. Nevertheless, I have chosen not to report the paper as LLM-written -- and maintain that choice -- because I am quite uncertain of this, and would rather give a chance to the authors rather than to 'blindly' dismiss their work. Once again, I profoundly apologize if this comment was undue. \
> Either way, the much bigger issue is the lack of sound mathematical reasoning, which still stands.
>
> **Equation (8):** ah yes, I'm not sure why I didn't see it the first time. All fine there. (Also, I don't think there is a need to add an introduction to multivariate derivatives to your submission just because of my slight :) )
>
> **Justification for lack of need of more experiments:** given that the theoretical component of the paper -- which the authors claim a number of times in their rebuttal is their main contribution -- is still rather shaky, I don't think it justifies a weaker experimental analysis. Moreover, adding such experiments can only serve to make the paper stronger, and I urge the authors to consider it for a future revision.
>
> **Equivalence between the estimators:** I understand that the authors claim there to be some sort of approximate equivalence between the estimators. I am still unconvinced; if you believe there is one, I suggest to prove it formally, and add it in the form of an additional proposition to the paper (even if just in the supplementary material). I would also very much like to see the experiments with the other estimator, even if the authors claim it to be worse. I will try soon to give one more read to the rebuttal given by the authors on this point to see if I missed something.
>
> Given the issues (especially the soundness one, and precisely in the spot that the authors claim is their main contribution), I maintain my score.

---

> > ### Author Response · Authors · 2024-11-27
> >
> > Please refer below for detailed responses to your concers:
> >
> > **Reviewer:** Derivatives that do not exist: the justification given in the rebuttal is very unsatisfactory (and the new appendix B.1 in the paper is not much better).
> >
> > **Authors:** Please specify what concrete aspects of our mathematical proof are problematic. Simply labeling things as "unsatisfactory" and "not much better" is not helpful in addressing technical concerns the reviewer might have.
> >
> > **Reviewer:** First, continuity and iid do not imply that ties happen with probability 0.
> >
> > **Authors:**  We disagree. We believe that the reviewer will concur with us that if we draw $n$ continuous random variables independently, the probability of two of them being identical is zero. The reviewer can refer, for example, to Chapter 10.6 on Kolmogorov-Smirnov test in [R2]: *Probability and Statistic* by DeGroot and Schervish, where the authors state "Since the observations come from a continuous distribution, there is probability 0 that any two of the observed values $x_1, ..., x_n$ will be equal." Note that, as explicitly stated in Proposition 3.1, $X$ is assumed to be an absolutely continuous random vector throughout the paper. We further remarked this fact in the revised version of the paper, in blue, adding a sentence in the problem formulation section. This allows us to establish that $E_\theta(X_1,Y_1),\ldots,E_\theta(X_n,Y_n)$ are iid absolutely continuous random variables. Let us now establish that, in general, if $E_1,\ldots,E_n$ are absolutely continuous independent random variables, then the probability of there being ties in $E_1,\ldots,E_n$ is zero. Indeed, first note that, for $i \neq j$ and denoting $p_{E_j}(e)$ the probability density function (PDF) of $E_j$, we have
> > \begin{align*}
> >     \mathbb{P}(E_i = E_j) &= \int_{-\infty}^{+\infty}\mathbb{P}(E_i = E_j\,|\,E_j = e) p_{E_j}(e)\\mathrm{d}e\\
> >     &= \int_{-\infty}^{+\infty}\mathbb{P}(E_i = e\,|\,E_j = e) p_{E_j}(e)\\mathrm{d}e\\
> >     &= \int_{-\infty}^{+\infty}\mathbb{P}(E_i = e) p_{E_j}(e)\\mathrm{d}e
> > \end{align*}
> > where the last equality follows from the assumed independence. Lastly, noting that $\mathbb{P}(E_i = e) = 0$ for any $e\in\mathbb{R}$ (which follows due to $E_i$ being a continuous random variable), then $\mathbb{P}(E_i = E_j) = 0$. Lastly, the probability of there being ties in $E_1,\ldots,E_n$ is
> > \begin{equation*}
> >     \mathbb{P}\left(\bigcup_{(i,j): i\neq j} \{(E_i = E_j)\}\right) \leq \sum_{(i,j): i\neq j}\mathbb{P}(E_i = E_j) = 0.
> > \end{equation*}
> >
> > [R2]: DeGroot, M. H., \& Schervish, M. J. (2011). *Probability and Statistics* (4th ed.). Pearson. (freely available on \href{https://github.com/muditbac/Reading/blob/master/math/Morris%20H%20DeGroot_%20Mark%20J%20Schervish-Probability%20and%20statistics-Pearson%20Education%20%20(2012).pdf}{GitHub}).

---

> ### Author Response · Authors · 2024-11-27
>
> **Reviewer:**  Derivative that do not exist: almost sure differentiability is not enough to guarantee the validity of the chain rule, the key property used in the derivation in the paper.
>
> **Authors:**
> We disagree. First note that in any neural network with ReLU activations the loss function is almost surely differentiable, and the chain rule is routinely and reliably applied. We will now provide some additional technical details to further support our claims on almost sure differentiability. The order statistics $E_{(1)}(\theta), \ldots, E_{(n)}(\theta)$ can be rewritten as $\big(E_{(1)}(\theta), \ldots, E_{(n)}(\theta)\big) = \mathbf{s}(E_\theta(X_1,Y_1),\ldots,E_\theta(X_n,Y_n))$, where $\mathbf{s}:\mathbb{R}^n \to \mathbb{R}^n$ denotes the sort function, mapping $(e_1,\ldots,e_n) \in \mathbb{R}^n$ to the unique point $(e_{(1)}, \ldots, e_{(n)})$ in $\mathbb{R}^n$ given by permuting $e_1,\ldots,e_n$ in such a way that $e_{(1)} \leq \ldots \leq e_{(n)}$. It is not difficult to see that $\mathbf{s}$ is piecewise linear and thus differentiable almost everywhere. In fact, it will be differentiable everywhere except in the finite union of hyperplanes $\{(e_1,\ldots,e_n): e_i = e_j\}$ with $i\neq j$. Let $A_n$ denote that finite union. It is not difficult to confirm that, outside $A_n$, the Jacobian matrix of $\mathbf{s}$ is simply given by a permutation matrix corresponding to the sorting permutation (which is unique when $(e_1,\ldots,e_n) \notin A_n$). More concretely, if $\mathbf{s}(\mathbf{e}) = (e_{\omega_1(\mathbf{e})}, \ldots, e_{\omega_n(\mathbf{e})})$, where $\mathbf{e} = (e_1,\ldots,e_n)$ and $\omega = (\omega_1(\mathbf{e}),\ldots,\omega_n(\mathbf{e}))$ denotes the unique sorting permutation for $\mathbf{e} \notin A_n$, then the Jacobian matrix of $\mathbf{s}$ evaluated at $\mathbf{e}\notin A_n$ is equal to a matrix where the $j$-th row is equal to the $\omega_j(\mathbf{e})$-th row of the $n\times n$ identity matrix. Let $u_1,\ldots,u_n$ denote the columns of the $n\times n$ identity matrix. Then,
> \begin{equation}
>     \frac{\partial\mathbf{s}}{\partial\mathbf{e}}(\mathbf{e}) = \begin{bmatrix}
>         u_{\omega_1(\mathbf{e})}^T\\
>         \vdots
>         u_{\omega_n(\mathbf{e})}^T
>     \end{bmatrix},
> \end{equation}
> (***Note***: $[ a_1 \vdots a_n]$ denotes a matrix with rows $a_1, ... , a_n$)
>
> (see, e.g., Blondel, Mathieu, et al. "Fast differentiable sorting and ranking." ICML, 2020.) and thus we can expect the chain rule to be of the form
> \begin{equation}
>     \frac{\partial}{\partial\theta} \begin{bmatrix}
>         E_{(1)}(\theta)\\
>         \vdots
>         E_{(n)}(\theta)
>     \end{bmatrix}
>     = \begin{bmatrix}
>         u_{\omega_1(\theta)}^T\\
>         \vdots\\
>         u_{\omega_n(\theta)}^T
>     \end{bmatrix}\begin{bmatrix}
>         \frac{\partial E}{\partial\theta}(\theta,X_1,Y_1)\\
>         \vdots\\
>         \frac{\partial E}{\partial\theta}(\theta,X_n,Y_n)
>     \end{bmatrix}
> \end{equation}
> where $\omega_j(\theta)$ is shorthand for $\omega_j(E_\theta(X_1,Y_1), \ldots, E_\theta(X_n,Y_n))$. Naturally, equation above holds almost surely. Let us confirm this formally. First, let $B_n(\theta)$ denote the event in which there are no ties in $E_\theta(X_1,Y_1), \ldots, E_\theta(X_n,Y_n)$, i.e. that $(E_\theta(X_1,Y_1), \ldots, E_\theta(X_n,Y_n))$ lies outside of $A_n$. As discussed earlier, $\mathbb{P}(B_n(\theta)) = 1$ for any $\theta$. Formally, the random vector $X$ and random variable $Y$ live in some probability space with some outcome space $\Omega$. Let $\xi\in\Omega$ denote a concrete outcome (we will avoid using the more common notation $\omega$ as it conflicts with previous notation). Now, if $\xi\in B_n(\theta)$, then $(E_{(1)}(\theta), \ldots, E_{(n)}(\theta)) = \mathbf{s}(E_\theta(x_1,y_1), \ldots, E_\theta(x_n,y_n))$ with $x_i = X_i(\xi)$ and $y_i = Y_i(\xi)$. Once we fix a concrete outcome, then $\theta \mapsto (E_{(1)}(\theta), \ldots, E_{(n)}(\theta))$ is just a regular function $\mathbb{R}^p \to \mathbb{R}^n$, where $p = \dim(\theta)$, and thus the usual notions of differentiability and chain rule from multivariate calculus are applicable. Let us fix $\theta = \bar{\theta}$. Following the argument in appendix B.1, where we note that the complement of $A_n$ is an open set, then, for every $\theta$ in some open neighborhood around $\bar{\theta}$, we have
> \begin{equation}
>     E_{(j)}(\theta) = E(\theta, x_{\omega_j(\bar{\theta})}, y_{\omega_j(\bar{\theta})}),
> \end{equation}
> where $\omega_j(\bar{\theta}) = \omega_j(E_{\bar{\theta}}(x_1,y_1), \ldots, E_{\bar{\theta}}(x_n,y_n))$ is constant in $\theta$. Clearly then $E_{(j)}(\theta)$ is continuously differentiable in $\theta$, with Jacobian matrix exactly as given in the chain rule equation above for outcome $\xi$, i.e. $X_i = x_i$ and $Y_i = y_i$. To conclude, note that, since $\xi \in B_n(\bar{\theta})$ and $\bar{\theta}$ were arbitrary, with $\mathbb{P}(B_n(\bar{\theta})) = 1$, it follows that the chain rule in the above equation holds almost surely.

---

> > ### Author Response · Authors · 2024-11-27
> >
> > **Reviewer:**
> > Now, I'll be honest, and please forgive me if I go too overboard or am incorrect here: reading this part of the rebuttal felt almost like reading mathematical nonsense. The reasoning was very reminiscent of the usual outputs of LLMs when doing mathematical proofs, in that at the same time it seemed very flawed, but with just a tint of plausibility that forced me not to dismiss it outright. And, to be honest, many bits of the original submission already gave me a slight LLM ick. Nevertheless, I have chosen not to report the paper as LLM-written -- and maintain that choice -- because I am quite uncertain of this, and would rather give a chance to the authors rather than to 'blindly' dismiss their work. Once again, I profoundly apologize if this comment was undue.
> >
> > **Authors:**
> > No LLM was used to write any part of paper, response or mathematical proof. If the reviewer has any specific and constructive suggestions for improvements, they are welcome.
> >
> >
> > **Reviewer:**
> > Justification for lack of need of more experiments: given that the theoretical component of the paper -- which the authors claim a number of times in their rebuttal is their main contribution -- is still rather shaky, I don't think it justifies a weaker experimental analysis. Moreover, adding such experiments can only serve to make the paper stronger, and I urge the authors to consider it for a future revision.
> >
> > **Authors:** Please note that we concur with the reviewer that more experiments would be valuable and indeed we plan to work on them as part of our future research. However, we believe that our main contribution is in identifying the sample-inefficiency in ConfTr *and* in providing an effective solution, with a fair comparison with ConfTr that clearly shows the superiority of our novel algorithm. In this sense, we believe our work is of high value to the community. Large-scale experiments are the natural next step - for both ConfTr and VR-ConfTr.
> >
> >
> > **Reviewer:**
> > Equivalence between the estimators: I understand that the authors claim there to be some sort of approximate equivalence between the estimators. I am still unconvinced; if you believe there is one, I suggest to prove it formally, and add it in the form of an additional proposition to the paper (even if just in the supplementary material). I would also very much like to see the experiments with the other estimator, even if the authors claim it to be worse. I will try soon to give one more read to the rebuttal given by the authors on this point to see if I missed something.
> >
> >
> > **Authors:**
> > Please note that, given a calibration batch and a parameter $\theta$, if we fix $\varepsilon$ as the smallest value such that $m$ conformity scores are $\varepsilon$-close to $\hat\tau(\theta)$, then the $m$-ranking estimator is exactly the same as the $\varepsilon$-threshold estimator. Using the $m$-ranking estimator, we allow $\varepsilon$ to change at each iteration, and we have empirically seen that this strategy is easier to tune (hyper-parameter-wise) and thus more effective - relative to trying to find a single value of $\varepsilon$ for all iterations - but the estimator is essentially an $\varepsilon$-threshold estimator, with $\varepsilon$ a function of the batch, of $m$ and of the parameter $\theta$. We could say that the $m$-ranking strategy is an heuristic way to tune $\varepsilon$ at each iteration when using an $\varepsilon$-threshold estimator. We added this remark when introducing the $m$-ranking estimator in the revised paper (in blue).

---

> ### Comment · Reviewer_eSTW · 2024-12-03
>
> **Ties with probability zero:** if we assume that the distribution is continuous (which was apparently being implicitly done), then indeed it is true. Okay on this bit, though I advise the authors to be more explicit about such assumptions in the future.
>
> **Almost sure differentiability is not enough:** I stand by this point. First, the argument of "note that in any neural network with ReLU activations the loss function is almost surely differentiable, and the chain rule is routinely and reliably applied": this does not mean that it is because it is 'almost everywhere defined'. The typical explanation uses subgradients, which I pointed towards in my original review. As for the following, more in depth explanation: again, the authors are being too lax with 'almost everywhere' / 'almost certainly'. I see no justification that, e.g., the E function does not end up mapping a lot of points to the set that has measure zero, where the function is not differentiable. By this point, it's not a matter of whether this can be shown -- the fact that after so many revisions I am still finding soundness issues in the authors' proofs is worrying.
>
> **Experiments:** the authors claim that "our main contribution is in identifying the sample-inefficiency in ConfTr and in providing an effective solution, with a fair comparison with ConfTr that clearly shows the superiority of our novel algorithm". However, this needs to be validated, through sound theory and/or substantial experiments. The theory does not seem sound -- and even if it was, it is hardly extensive -- and the experiments, while they would have been in decent quantity were the theory substantial, it is not, and so more would be required to validate the approach.
>
> Given my substantial concerns, I maintain my score of rejection.

---

> > ### Author Response · Authors · 2024-12-04
> >
> > Please refer below for our detailed responses to your most recent concerns:
> >
> > **Reviewer:** Ties with probability zero: if we assume that the distribution is continuous (which was apparently being implicitly done), then indeed it is true. Okay on this bit, though I advise the authors to be more explicit about such assumptions in the future.
> >
> > **Authors:**
> > We would like to point out again that we had originally already explicitly stated in Proposition 3.1 that X is assumed to be an absolutely continuous random vector. We further remarked this fact in the revised version of the paper (note: this was already done in the first round of revision) in
> > blue, adding a sentence in the problem formulation section, after your first comments with concerns on equation (5).
> >
> > We would also like to raise attention to the fact that assuming input features to be continuous random variables is an extremely common practice in machine learning and statistical analysis.
> >
> > **Reviewer:** Almost sure differentiability is not enough: I stand by this point. First, the argument of "note that in any neural network with ReLU activations the loss function is almost surely differentiable, and the chain rule is routinely and reliably applied": this does not mean that it is because it is 'almost everywhere defined'. The typical explanation uses subgradients, which I pointed towards in my original review. As for the following, more in depth explanation: again, the authors are being too lax with 'almost everywhere' / 'almost certainly'. **I see no justification that, e.g., the E function does not end up mapping a lot of points to the set that has measure zero, where the function is not differentiable.** By this point, it's not a matter of whether this can be shown -- the fact that after so many revisions I am still finding soundness issues in the authors' proofs is worrying.
> >
> > **Authors:** Please note that, E is by definition a continuous random variable (being a continuous function of a continuous random variable).
> > This alone is enough to guarantee the absence of ties in $E_1, ..., E_n$ with probability one, which is used in the appendix to rigorously confirm the differentiability with probability one of the ordered statistics $E_{(1)}(\theta), ..., E_{(n)}(\theta)$.
> > Hence, what the reviewer is asserting about mappings to sets of measure zero is not pertinent. We would like to remark that what we just wrote is not a proof, but just well-known and basic facts in probability theory and calculus. Therefore, we do not see any supportive argument against the soundness of our paper.
> >
> >
> > **Reviewer:** Experiments: the authors claim that "our main contribution is in identifying the sample-inefficiency in ConfTr and in providing an effective solution, with a fair comparison with ConfTr that clearly shows the superiority of our novel algorithm". However, this needs to be validated, through sound theory and/or substantial experiments. The theory does not seem sound -- and even if it was, it is hardly extensive -- and the experiments, while they would have been in decent quantity were the theory substantial, it is not, and so more would be required to validate the approach.
> >
> > **Authors:** We would like to note that we have rigorously addressed all concerns the reviewer has raised about our paper. We believe every bit of mathematical analysis in the paper is correct, and that the theory is sound. Addressing the concerns raised by the reviewer just **consolidated our confidence** in our results, and we are disappointed to hear the reviewer still has concerns about the soundness of our mathematical reasoning and analysis, after the extensive and rigorous explanations we have provided both here in the responses and in the appendix of the revised paper.

---

### Official Review · Reviewer_vKbW · 2024-11-04

**Soundness:** 2
**Presentation:** 2
**Contribution:** 2
**Rating:** 5
**Confidence:** 3

**Summary:**

The submission proposes VR-ConfTr, a variance-reduced conformal training method that enhances the stability and efficiency of conformal prediction by reducing the variance in gradient estimates, which is a key limitation in previous [1]. VR-ConfTr introduces a novel gradient estimator that achieves faster convergence and smaller prediction sets.

---

[1] Stutz, David, et al. "Learning Optimal Conformal Classifiers." International Conference on Learning Representations.

**Strengths:**

- The length efficiency in conformal prediction is an important research problem. The proposed novel approach addresses significant sample inefficiency issues present in prior methods and stabilizes gradient estimation.

- This is a plug-in algorithm hence the proposed variance reduction technique could enjoy a broad applicability.

- VR-ConfTr demonstrates superior performance across multiple datasets, including MNIST, Fashion-MNIST, Kuzushiji-MNIST, and OrganAMNIST.

**Weaknesses:**

- The proposed VR-ConfTr gradient estimation method uses sample splitting, where batch data are divided into calibration and prediction subsets. This approach might introduce sampling instability, particularly with smaller datasets or non-i.i.d. data. This limitation is not sufficiently analyzed in the paper, nor do the authors consider alternative sampling strategies that might enhance stability.

- Although the paper claims innovation in integrating variance reduction into conformal training, similar variance reduction techniques have already been proposed in gradient estimation contexts (e.g. well-known SVRG in optimization community). The authors fail to convincingly differentiate their work from these established methods, nor do they address why VR-ConfTr is distinct beyond the context of conformal prediction.

- The empirical results, while presented across multiple datasets, appear selective and limited to low-complexity, classical datasets (e.g., MNIST variants). Conformal prediction applications span far more complex domains, yet there is no evidence here that VR-ConfTr scales or performs effectively on realistic, large-scale datasets.


- The theoretical section of this paper appears hastily constructed (see Questions), which risks undermining the reader's confidence in the proposed methodology and suggests that this submission is not yet ready for formal publication.

**Questions:**

- The mathematical notation in Theorem 3.1 lacks standardization. Are you intending to use $q_{\epsilon}(\theta)^n$ from (i) and $
[q\_{\epsilon}(\theta)]^n$ from (ii) to both represent the $n$-th power of $q(\cdot)$?

- line 323: the definition of $K(t)$ lacks a right parenthesis

- line 716, the asymptotic equivalence is defined by $a_n \asymp b_n: \lim_{n \to \infty} \frac{a_n}{b_n} = 1$ (in your sense), so we should have $b_n \neq 0$, which is in contrast to your usage like line 719, 745. Besides, "Let"

- line 852: missing a proper definition

- if you let $a_n=nf_n$, shouldn't your (45) turns into $a_{n+1}=(n+1)q f_n+(1-q^n)$ instead of your (46)?

- In my view, the notation in the second line of (50) should begin with $ \preceq$. Also the = in the final line seems unclear to me. While I understand your intention to retain the $\frac{2 - p}{pn} \Sigma_\epsilon$, the remaining terms are somehow ambiguous—how exactly does it equate to your final conclusion?

---

> ### Author Response · Authors · 2024-11-22
>
> Thank you for your review, we are happy to address your concerns in the following.
>
> **Reviewer**:
> 1. The proposed VR-ConfTr gradient estimation method uses sample splitting, where batch data are divided into calibration and prediction subsets. This approach might introduce sampling instability, particularly with smaller datasets or non-i.i.d. data. This limitation is not sufficiently analyzed in the paper, nor do the authors consider alternative sampling strategies that might enhance stability.
>
> **Authors**:
>
> Please note that the sampling instability in the ConfTr algorithm is *precisely what our novel proposed  plug-in algorithm (VR-ConfTr) addresses*. In order to simulate conformal prediction during training, ConfTr needs to estimate the population $\alpha$-quantile ${\tau}(\theta)$ of the conformity scores. Splitting the batch into calibration and prediction is key to estimate $\hat{\tau}(\theta)$ during training. A poor estimate of  ${\tau}(\theta)$ and its gradient $\frac{\partial{{\tau}}}{{\partial \theta}}(\theta)$  hinders learning.  As we show as one of the main contributions of our paper, (ConfTr) falls short of providing a good estimate of $\frac{\partial{{\tau}}}{{\partial \theta}}(\theta)$ and thus utilizes the calibration data ***inefficiently***. With our proposed VR-ConfTr algorithm, thanks to our plug-in technique and a variance reduced estimate of $\frac{\partial{{\tau}}}{{\partial \theta}}(\theta)$,
>  we provably address this issue, remarkably *without requiring a larger calibration batch and at essentially the same computational cost as* ConfTr. In other words, we design an algorithm which is *sample-efficient* and effectively deals with the instability problem of ConfTr.

---

> > ### Author Response · Authors · 2024-11-22
> >
> > **Reviewer:**
> >
> > 2. Although the paper claims innovation in integrating variance reduction into conformal training, similar variance reduction techniques have already been proposed in gradient estimation contexts (e.g. well-known SVRG in optimization community). The authors fail to convincingly differentiate their work from these established methods, nor do they address why VR-ConfTr is distinct beyond the context of conformal prediction.
> >
> > **Authors:**
> >
> > Please note that our VR-ConfTr method is specifically designed for the ConfTr objective function, which is in turn designed to optimize a model according to conformal prediction efficiency metrics. As outlined by our novel analysis, the existing ConfTr algorithm has a clear source of sample inefficiency, which pushes us to design and validate a tailored solution. The source of sample inefficiency of ConfTr specifically comes from simulating conformal prediction in-between training updates, which introduces the need of estimating and using the gradient of a quantile function, $\frac{\partial \tau}{\partial \theta}(\theta)$. Note that this is a specific feature of conformal training and it is not present in classic optimization and machine learning problems. In this sense, we note that our technique is fundamentally different from existing variance reduction techniques in optimization, like SVRG. In particular, we note that SVRG is based on the idea of *increasing the number of samples used in the estimate of the gradient to reduce the variance*, which is done by periodically computing a full gradient (using the full dataset) to perform the updates. Hence, in SVRG, variance is reduced increasing (wisely) the number of samples used to estimate gradients. As another example, SAGA is another well-known approach for variance reduction in optimization, which stores past gradients to again increase the number of samples used at each iteration to reduce the variance of the updates. For both SVRG and SAGA, we thus note that they achieve variance reduction by (i) increasing the number of samples used at each iteration, (ii) increasing the computational effort and (iii) increasing the memory usage. This is a crucial conceptual difference with respect to our CRM-tailored approach, as we note that our technique *does not require increasing the batch size, nor the memory usage, and it has essentially the same computational cost of the existing ConfTr algorithm*. Our proposed VR-ConfTr is indeed based on plugging an improved estimate of a specific quantity - the population quantile gradient $\frac{\partial \tau}{\partial \theta}(\theta)$ - in the conformal training cost function gradient *making better use of the data samples in the calibration batch*. Note that the computational complexity of our variance-reduced algorithm is essentially the same as ConfTr, and we are using the *exact same batch size* (and memory) as ConfTr. Our variance reduction approach can be seen as a novel way to be *sample-efficient* in conformal training, something that the existing ConfTr algorithm was failing to do. Finally, note that, for how it is designed, VR-ConfTr does not prevent the use of classic variance reduction techniques (including, potentially, SVRG and SAGA themselves) on top of it to further improve performance. This is indeed potential material for future investigations.

---

> > > ### Author Response · Authors · 2024-11-22
> > >
> > > **Reviewer:**
> > >
> > > 3. The empirical results, while presented across multiple datasets, appear selective and limited to low-complexity, classical datasets (e.g., MNIST variants). Conformal prediction applications span far more complex domains, yet there is no evidence here that VR-ConfTr scales or performs effectively on realistic, large-scale datasets.
> > >
> > > **Authors:**
> > > With respect to larger scale datasets, note that our key contribution in this paper is to detect and provide an effective solution to the sample inefficiency problem of the original ConfTr algorithm. This is a critical problem ***even with classical datasets and relatively small models***. We believe that our work is therefore a significant first step towards the possibility of applying conformal training techniques to large scale models and datasets. The experiments with the medical dataset OrganA-MNIST and the clear superior performance and effectiveness of VR-ConfTr as compared to ConfTr  with this relatively more complex dataset are promising in this direction. We believe that hyper-parameter tuning needed for higher resolution datasets and large-scale models necessitates extensive investigation and it is beyond the scope of this paper.

---

> ### Author Response · Authors · 2024-11-22
> **Minor issues**
>
> Thank you for all your questions related to notation and minor issues in the paper. We have addressed all your points in the revised paper - both in the main paper and in the appendix.
>
> With respect to $a_{n+1}$, note that the equation written by the reviewer becomes precisely our (46) because $a_n = nf_n$
>
>
> We updated (50) using $\preceq$ as suggested by the reviewer, and added two steps clarifying the final relation that is indeed $\preceq$

---

> > ### Author Response · Authors · 2024-11-26
> > **Follow-up request**
> >
> > Dear reviewer, thank you again for your review. Please let us know if you have any follow-up questions or comments, we would be happy to address any further concern or question you might have on our paper and responses.

---

### Official Review · Reviewer_YZv9 · 2024-11-08

**Soundness:** 3
**Presentation:** 2
**Contribution:** 2
**Rating:** 5
**Confidence:** 3

**Summary:**

In this work, authors studies conformal training and proposes variance-reduced conformal training which incorporates a variance reduction technique in the gradient estimation of the ConfTr objective function. In particular, it proposes an $\epsilon$ estimator to estimate quantile gradient. Authors also provide the theoretical results of the variance and bias of the proposed estimator, and some numerical studies on synthetic datasets and MNIST dataset.

**Strengths:**

1. Authors state the problem clearly and then propose a solution, finally provide numerical studies.
2. The discussion is accompanied with sufficient background introduction
3. Authors tackle the high-variance issue in conformal training  and reduce the variance of quantile estimator relaxing the definition of the gradient and using more available samples in estimation.

**Weaknesses:**

1. Some of the formulas are not clearly explained. In particular, what does the set ${E_{\theta}(X,Y) =\tau_{\tau}}$ represent? It seems $E_{\theta}(X,Y)$ is a random variable, $\tau_{\tau}$ is a scaler, but how to interpret it in ${E_{\theta}(X,Y) =\tau_{\tau}}$? Furthermore, what expectation you take with respect to in equation 11 and 12?
2. How do you the value of $\epsilon$? Can you include some ablation studies to this hyper-parameter?
3. What is the per-step computational cost of your algorithm compared to the ConfTr?
4. Can you test your method over larger scale datasets beyond the toy dataset of MNIST?

**Questions:**

Please see the weakness above.

---

> ### Author Response · Authors · 2024-11-22
>
> Thank you for your review. We are happy to respond to your concerns. We provide our point to point responses below:
>
> **Reviewer**:
> 1. Some of the formulas are not clearly explained. In particular, what does the set ${E_{\theta}(X,Y) =\tau_{\tau}}$ represent? It seems $E_{\theta}(X,Y)$ is a random variable, $\tau_{\tau}$ is a scaler, but how to interpret it in ${E_{\theta}(X,Y) =\tau_{\tau}}$? Furthermore, what expectation you take with respect to in equation 11 and 12?
>
> **Authors:**
>
> Please note that $E_{\theta}(X, Y) = E(\theta, X, Y)$ is a (scalar) function (conformity score) of the sample $(X, Y)$ - where $X$ is the feature vector and $Y$ is the corresponding label - and of the parameter $\theta$. The event $E_{\theta}(X, Y) = \tau(\theta)$ means that the conformity score function $E_{\theta}(X, Y)$ is precisely equal to the population quantile $\tau(\theta)$. The expectations in equations (11) and (12) are taken with respect to the distribution of the sample $(X, Y)$, which is the only random object in the argument of the expectations, that is the function $\frac{\partial E}{\partial \theta}(\theta, X, Y)$, because we consider a fixed $\theta$ (as we do in Theorem 1). Please let us know if this is sufficient to clarify your concerns.
>
> **Reviewer:**
>
> 2. How do you the value of $\epsilon$? Can you include some ablation studies to this hyper-parameter?
>
> **Authors:**
>
> In light of the reviewer's question, we have added section C.2 in the appendix, conducting an ablation study for the $\varepsilon$-estimator on the GMM dataset. We report both the bias and variance of the estimated quantile gradients for each different value of $\varepsilon$.
> Moreover, in section C.2 we explore the intimate connection between the $m$-ranking and the $\varepsilon$-threshold estimators both theoretically and empirically. We first note that a good choice of $\varepsilon$ changes significantly across iterations making it hard to find a single good hyper-parameter $\varepsilon$. A potential choice to effectively adapt $\varepsilon$ during training is the $m$-ranking estimator. The $m$-ranking estimator can be seen as a special case of the $\varepsilon$-estimator in which $\varepsilon$ is chosen as the smallest value for which $m$ samples' conformity scores fall within $\varepsilon$-distance from $\hat{\tau}(\theta)$.
> To empirically illustrate this connection, we have included an additional study validating the importance of dynamically tuning $\varepsilon$, demonstrating how the threshold $\varepsilon$ containing $m$ conformity scores significantly changes across epochs.
> We hope section C.2 in the appendix fully addresses the reviewer's question.
>
>
> **Reviewer:**
>
> 3. What is the per-step computational cost of your algorithm compared to the ConfTr?
>
> **Authors:**
>
> The per-step computational cost of our proposed VR-ConfTr is essentially the same as ConfTr. In fact, our proposed "plug-in" technique consists in replacing a specific component in the gradient estimation pipeline of the ConfTr algorithm, which is the estimation of the population quantile gradient, $\frac{\partial \tau}{\partial \theta}(\theta)$. Indeed, by differentiating the objective directly as a function of the sample quantile $\hat{\tau}(\theta)$, ConfTr ends up using the gradient of the sample quantile $\frac{\partial \hat{\tau}}{\partial \theta}(\theta)$. As one of the main contributions of this paper, we show that this specific estimator is sample inefficient, and our proposed VR-ConfTr is specifically designed to replace it - thanks to our ``plug-in" technique - with a better, provably sample-efficient estimator $\widehat{\frac{\partial {\tau}}{\partial \theta}}$. In particular, we adopt the $m$-ranking estimator, which can be seen as a special case of the $\epsilon$-threshold estimator, which, as we show analytically, provably reduces the estimation variance with the calibration batch size, i.e., it is sample-efficient. Remarkably, the computational complexity of this estimator is essentially the same as computing $\frac{\partial \hat{\tau}}{\partial \theta}(\theta)$ - which is what is done in ConfTr - because it just boils down to computing and averaging $m$ derivatives of conformity scores. The other parts in the estimation of the objective function gradient have exactly the same computational complexity for both algorithms, and thus we can conclude that the computational complexity of the two algorithms is essentially the same.
> We provide a more detailed description of this reasoning in a new dedicated Appendix E in the revised version of the paper.

---

> > ### Author Response · Authors · 2024-11-22
> >
> > **Reviewer:**
> >
> > 4. Can you test your method over larger scale datasets beyond the toy dataset of MNIST?
> >
> > **Authors:**
> >
> > We note that we tested our approach on four widely adopted datasets: MNIST, Fashion-MNIST, Kuzushiji-MNIST, and OrganAMNIST. Although all datasets have similar input dimensions, they are of increasing levels of difficulty. We note that both Kuzushiji-MNIST and OrganAMNIST were not included in the original ConfTr experiments. Also note that OrganAMNIST is a real-world medical benchmark comprising axial CT slices of the abdominal region, and it poses significant challenges.
> > With respect to larger scale datasets, note that our key contribution in this paper is to detect and provide an effective solution to the sample inefficiency problem of the original ConfTr algorithm. This was a critical problem ***even with benchmark datasets and relatively small models***. We believe that our work is therefore a significant first step towards the possibility of applying conformal training techniques to large scale models and datasets. The experiments with the medical dataset OrganA-MNIST and the clear superior performance and effectiveness of VR-ConfTr as compared to ConfTr  with this relatively more complex dataset is promising in this direction.

---

> > ### Author Response · Authors · 2024-11-26
> > **Follow-up request**
> >
> > Dear reviewer, thank you again for your review. Please let us know if you have any follow-up questions or comments, we would be happy to address any further concern or question you might have on our paper and responses.

---

> > > ### Author Response · Authors · 2024-11-27
> > > **Follow-up**
> > >
> > > Dear reviewer, thank you again for your review. Given that this is the last day to update the revised manuscript, we would like to take the opportunity to ask you for further follow-up comments or questions that might help us improve the paper.
> > > We are also happy to elaborate more in the upcoming discussion days for any other queries or comments you might have on paper and responses.

---

> > > > ### Author Response · Authors · 2024-12-02
> > > > **Final follow-up**
> > > >
> > > > Dear reviewer, we greatly appreciate your feedback. Given that this is the last day for us to receive your responses, please let us know if we were able to address all your concerns.

---

### Author Response · Authors · 2024-11-25
**Message to reviewers**

Dear reviewers, thank you for your reviews, we have done our best to respond to all your questions and comments, and we hope you will positively evaluate our paper in light of our clarifications and responses.
Please let us know if you have any follow up comments or questions on your end, we will be very happy to respond before the discussion period ends.

---

### Author Response · Authors · 2024-12-04
**Final general comment**

As the end of the review period approaches, we would like to thank the reviewers for their time and feedback. We have diligently worked to address all your comments and have made corresponding revisions to the updated paper.
We would appreciate it if you could take a moment and revisit our revisions and responses to your comments. If you find our explanations and revisions satisfactory, we would greatly appreciate your consideration in updating your scores to reflect this.

We would like to note that, while we appreciate the time Reviewer eSTW has dedicated to the discussion period, we respectfully disagree with the soundness issues they have raised, and with several statements they have made during the discussion. We have extensively addressed reviewer eSTW's comments in our responses.

Thank you again for your time and consideration,

Sincerely,

Authors

---

### Meta-Review · Area_Chair_S7e1 · 2024-12-21

**Metareview:**

The paper proposes a novel method to improve the efficiency and stability of conformal training for classification tasks by addressing challenges in gradient estimation. More precisely, it introduces a novel variance reduction technique for estimating quantile gradients where authors highlight a core issue that there is no universal consensus on an estimator for the population quantile of scalar random variables along with sample inefficiency (the variance of the gradient of quantile estimates is approximately constant when the sample size is moderately large). This inefficiency propagates to the estimate of the gradient of the loss function.

The quantile estimator in Equation (4) by Hyndman & Fan (1996) forms the basis of their analysis. While this estimator is widely recognized, it is 28 years old, and more sophisticated methods have emerged since then (e.g., kernel-based, optimal transport based, and even using neural net parametrization along with pinball loss minimization etc ...).  The paper does not adequately justify why this particular estimator is chosen over more modern approaches. Does this proposed variance reduction method generalizes to other estimators?


I really do appreciate the analysis but was left unclear, typically the decoupling arising with the connexion with quantile sensitivity in Proposition 3.1 could be more detailed. The core contribution is provided in Equation 13, which is a carefully reweighing of the gradient of the score function, is pretty hard to follow and deserves a more detailed explanation and analysis. A clearer derivation of how this sensitivity is exploited practically in their variance-reduced estimator. More intuitive explanations or examples showcasing why this decoupling is advantageous.

The Variance reduction theorem seems to assume perfect quantile estimates which a bit strange no, even misleading? A reviewer mentioned it and I found the authors answer very unclear, and, in all case is an asymptotic argument. Using the same argument, if the quantile estimate is correct enough, what's the point of the paper then? The authors argue that  large calibration batch size is not enough to provide a sample efficient gradient estimation, observation. Ok, then what about large training size?

Also, once a (differentiable) loss function is defined and the problem is formulated as an Empirical Risk Minimization, there are indeed a wide variety of variance reduction techniques tailored for Stochastic Gradient Descent based algorithm. These techniques are well-studied and can directly improve the efficiency of gradient-based optimization methods. What is the issue with that?

I believe that the paper approach is very interesting but the current delivery is incomplete and the paper clarify should be improved.
Scores ranged from "marginally below acceptance" to "reject", with soundness issues and limited experiments being the major concerns.
Reviewers acknowledged the importance of the problem and the potential value of the method but emphasized the need for substantial revisions in both theory and experimentation. As such, I recommend a *reject* and encourage the authors to resubmit.

**Additional Comments On Reviewer Discussion:**

The overall discussions between the reviewers and authors did not foster a clear consensus to strictly reject the paper but were not supportive enough to accept the paper. I found the interactions with a particular reviewer on some technical mathematical details to lack the desired level of constructiveness and elegance. These issues appear largely fixable, and I encourage the authors to address them. However, I also acknowledge that adding such minor details may risk making the paper overly cumbersome.

- *Reviewer YZv9*
appreciated the clarity in the problem statement and the proposed solution's focus on addressing the high-variance issue in conformal training. They found the discussion on background concepts sufficient but raised concerns about the clarity of some key formulas and criticized the lack of ablation studies for the $\varepsilon$ estimator hyperparameter and noted that experiments were limited to small-scale datasets like MNIST. The reviewer also requested better explanations for the computational cost of VR-ConfTr compared to ConfTr.

- *Reviewer vKbW*
recognized the relevance of optimizing conformal prediction set sizes and noted that VR-ConfTr is a broadly applicable plug-in algorithm with performance improvements over ConfTr. However, they expressed concerns about the stability of the sampling process in VR-ConfTr, particularly when calibration and prediction subsets are small or non-i.i.d. They criticized the paper for not sufficiently differentiating its variance reduction approach from established methods like SVRG and highlighted the limited scope of the experiments, which focused primarily on MNIST-like datasets. The reviewer also pointed out ambiguities in the theoretical section.

- *Reviewer eSTW*
focus on improving conformal prediction efficiency interesting but raised significant concerns about the mathematical soundness of key derivations, particularly the validity of derivatives in Equation (5) and the use of the chain rule. They felt the theoretical analysis was overly simplified, failing to address general cases or alternative estimators comprehensively. The experiments, while promising, were deemed insufficient for validating the method’s scalability and generalizability. The reviewer emphasized the need for additional datasets and architectural diversity, including tests on transformers and high-resolution image tasks, to bolster the claims of the paper (Beyond CNN on mnist: ViTs, Transformers (e.g., BERT-style), GNNs for graph-based tasks etc ...

- *Reviewer YP6a*
praised VR-ConfTr for improving prediction efficiency and providing a theoretical analysis of variance reduction but noted its reliance on large calibration sets for accurate quantile estimation. They expressed concerns about potential computational overhead in resource-limited scenarios and the limited generality of the experiments, which focused on MNIST-type datasets. The reviewer encouraged the authors to evaluate VR-ConfTr on high-resolution images or larger datasets and to provide a more detailed discussion of its behavior with limited or imbalanced calibration data.

---

### Decision · Program_Chairs · 2025-01-22

Reject